# Transparent soil microcosms for live-cell imaging and non-destructive stable isotope probing of soil microorganisms

Kriti Sharma[1†], Márton Palatinszky[2], Georgi Nikolov[2], David Berry[2], Elizabeth A Shank[1,3,4]*

[1]Department of Biology, University of North Carolina, Chapel Hill, United States; [2]Department of Microbiology and Ecosystem Science, Centre for Microbiology and Environmental Systems Science, University of Vienna, Vienna, Austria; [3]Department of Microbiology and Immunology, University of North Carolina, Chapel Hill, United States; [4]Program in Systems Biology, University of Massachusetts Medical School, Worcester, United States

**Abstract** Microscale processes are critically important to soil ecology and biogeochemistry yet are difficult to study due to soil's opacity and complexity. To advance the study of soil processes, we constructed transparent soil microcosms that enable the visualization of microbes via fluorescence microscopy and the non-destructive measurement of microbial activity and carbon uptake in situ via Raman microspectroscopy. We assessed the polymer Nafion and the crystal cryolite as optically transparent soil substrates. We demonstrated that both substrates enable the growth, maintenance, and visualization of microbial cells in three dimensions over time, and are compatible with stable isotope probing using Raman. We applied this system to ascertain that after a dry-down/rewetting cycle, bacteria on and near dead fungal hyphae were more metabolically active than those far from hyphae. These data underscore the impact fungi have facilitating bacterial survival in fluctuating conditions and how these microcosms can yield insights into microscale microbial activities.

*For correspondence: Elizabeth.Shank@umassmed.edu

Present address: †Division of Geological and Planetary Sciences, California Institute of Technology, Pasadena, United States

## Introduction

Terrestrial soils are habitats to an unparalleled abundance and diversity of bacteria and fungi (*Delgado-Baquerizo et al., 2016*; *Fierer and Jackson, 2006*; *Horner-Devine et al., 2004*). The metabolic activities of these microbes drive critical biogeochemical processes with biosphere-level effects (*Pold and DeAngelis, 2013*; *Wieder et al., 2013*). However, the opacity and complexity of natural soils present a formidable challenge to the study of soil microbes in their native habitats.

'Transparent soils' (TS) are model systems where particles with a similar refractive index (RI) as their saturating liquids allow transmission of light and render a porous, 'soil-like' system optically transparent (*Iskander, 2010*). TS systems have been used for over 25 years in geoengineering and hydrology, where they have helped solve challenging problems in soil physics (*Iskander et al., 2015*). Despite their importance in that field, however, TS systems have only recently and rarely been used for applications in biology. In part, this is due to soil physicists favoring TS systems that use RI-matching liquids that do not support life, such as silicone oil. In 2005, however, work by *Leis et al., 2005* introduced the synthetic fluoropolymer Nafion (Chemours, Wilmington, DE) as a TS that could be RI-matched to water-based solutions that are compatible with culturing microorganisms in situ. *Downie et al., 2012*; *Downie et al., 2014* then demonstrated that plant roots with native-like architectures could be grown and visualized in Nafion-based TS (*Downie et al., 2012*). Surprisingly, given the novel visualization abilities it provides, the Nafion TS system has only been

sparsely utilized for studying biological systems (*Downie et al., 2014*; *Downie et al., 2012*; *O'Callaghan et al., 2018*). This may be due to (a) the high cost of Nafion and (b) the RI of Nafion (RI = 1.35; *Leis et al., 2005*) not matching water closely enough (RI = 1.333) to allow deep (millimeter to centimeter) imaging. Indeed, Ma and colleagues cited these reasons as their motivation for developing an inexpensive hydrogel-bead-based TS system for root phenotyping (*Ma et al., 2019*). These hydrogel beads, however, are poorly suited for microbial visualization due to their large size (~500 µm) (*Ma et al., 2019*), and their susceptibility to degradation by microbes as a carbon source (*Rice et al., 1992*; *Lin et al., 2018*).

Despite its limitations for plant root imaging studies, as we show here, Nafion remains a useful and versatile model for soil microbial ecology studies when microcosms utilizing only a few micrograms of TS are used; these chambers only require imaging depths of tens or hundreds of microns (and thus are tolerant of slight RI mismatches) and are inexpensive to build. We further show that cryolite (a naturally-occuring sodium aluminum fluoride crystal) is another promising biocompatible TS substrate with high potential for microbial studies. Cryolite's RI (=1.339) (*Lewis, 2007*) renders it effectively entirely transparent in aqueous solutions. Cryolite has previously been utilized to gain insights into the burrowing behaviors of marine invertebrates (*Flessa, 1972*; *Francoeur and Dorgan, 2014*; *Dorgan, 2018*), the swimming behavior of *Bacillus subtilis* (*Zhu et al., 2014*), and the spatial patterns of oxygen consumption by a bacterial pseudomonad species (*Oates et al., 2005*), but its potential for microbial ecology studies remains largely unexplored.

Gaining greater insight into microbial spatial distributions, migration, and growth dynamics – as well as into the physiological states and ecological functions of individual cells and species – is critical to the future of the field (*Fike et al., 2008*; *Berry et al., 2015*). In this study, we critically evaluate the capabilities of both Nafion and cryolite as TS substrates for advancing experimental research in soil microbial ecology by generating three-dimensional matrices containing pore spaces analogous to those bacteria inhabit in terrestrial soils (*Dal Ferro and Morari, 2015*; *Deng et al., 2015*; *Baveye et al., 2018*). We show that TS microcosms made of both Nafion and cryolite are amenable to high-resolution, three-dimensional imaging by fluorescence and confocal microscopy, and additionally are compatible with Raman microspectroscopy – a powerful non-destructive method to obtain physiological information about cell states and microbial metabolic activity and nutrient uptake (*Huang et al., 2004*; *Huang et al., 2009*; *Li et al., 2012*; *Li et al., 2013*; *Berry et al., 2015*; *Kumar B N et al., 2016*). Both TS substrates enable the measurement of deuterium uptake as a marker of microbial activity, while cryolite-based TS microcosms further enable the measurement of microbial uptake of isotopically labeled ($^{13}$C) carbon.

We use these tractable, structurally complex TS systems to address an important and experimentally challenging question in soil microbial ecology: how bacteria respond to desiccation and rehydration. Specifically, we ask how metabolically active bacteria are within the TS matrix depending on their proximity to fungal hyphae after a desiccation event. To our knowledge, this is the first study to perform Raman microspectroscopy of cells within a transparent porous matrix, with important potential applications to fundamental problems in soil and sediment microbial ecology. We anticipate that approaches described here will establish these TS systems as novel tools to non-destructively monitor microbial distributions and activity as well as carbon flow through complex, porous, soil-like systems to answer important questions about the ecophysiology of microbes within soils.

## Results

### Overview of TS microcosms

We used standard microfluidics procedures to generate a visualization chamber out of polydimethylsiloxane (PDMS), a non-toxic gas-permeable silicone polymer commonly used for microfluidics fabrication (*Figure 1A*; see Materials and methods). The chambers were designed as 3 × 5 mm hexagons (free of 90° corners that can produce regions of low mixing). Inlet and outlet channels were 150 µm wide and 3 mm long, and the inlet and outlet ports themselves were 1 mm circles. Chamber height was 100 µm. As detailed below, the TS matrices within these chambers were optically transparent (*Figure 1B*) and enabled the three-dimensional visualization of bacteria held within them (*Figure 1C*).

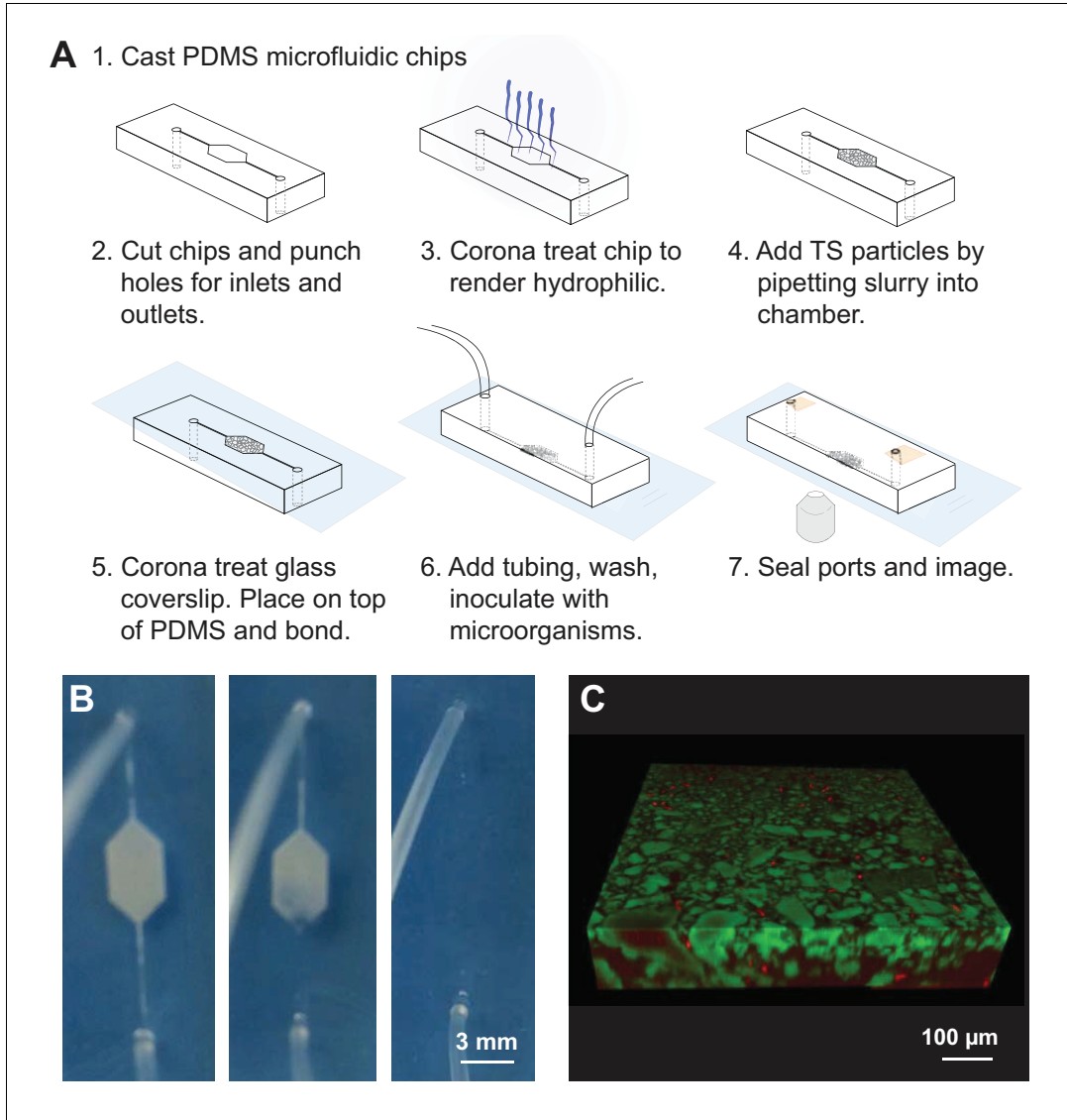

**Figure 1.** Transparent soil (TS) microcosms. (**A**) Manufacture process of microcosm fluidics chambers. (**B**) 20% ethanol added after chip manufacture hydrates dry, hydrophobic Nafion and renders it transparent. Microfluidics chamber (3 × 5 mm hexagon, with 200 μm wide channels) filled with Nafion and attached by tubing to syringe with 20% ethanol, held in syringe pump. As ethanol is slowly flowed into the microcosm by the syringe pump, the Nafion hydrates and becomes transparent. Rehydrated Nafion can then be washed with media, washing away ethanol and rendering microcosms suitable for cell culture. (**C**) Three-dimensional confocal rendering of fluorescently labeled *E. coli* cells visualized to 100 μm depth in Nafion-based TS microcosm by confocal microscopy. Sulforhodamine-stained Nafion particles (false-colored green), and *E. coli* cells constitutively expressing cyan fluorescent protein (P*spacC*-*cfp*, false-colored red). Scale bar = 100 μm.

## Visualization of TS matrices

To determine whether pore space could be adequately distinguished from particle space within our model soil systems, we first tested whether particle edges could be rendered visible in the two TS matrices. (Note that for speed and ease, these fluorescence parameterization experiments were conducted in microwell slides rather than in PDMS microcosms). Nafion particle edges were visible through water by brightfield or differential interference contrast (DIC) microscopy (*Figure 2A*). As shown previously (*Downie et al., 2012*), Nafion can be rendered fluorescent in the RFP channels by staining with sulforhodamine (*Figure 2A*). We found that a small proportion (≤1%) of Nafion particles were also slightly autofluorescent in the FITC channel (*Figure 2A*). For context, the detected

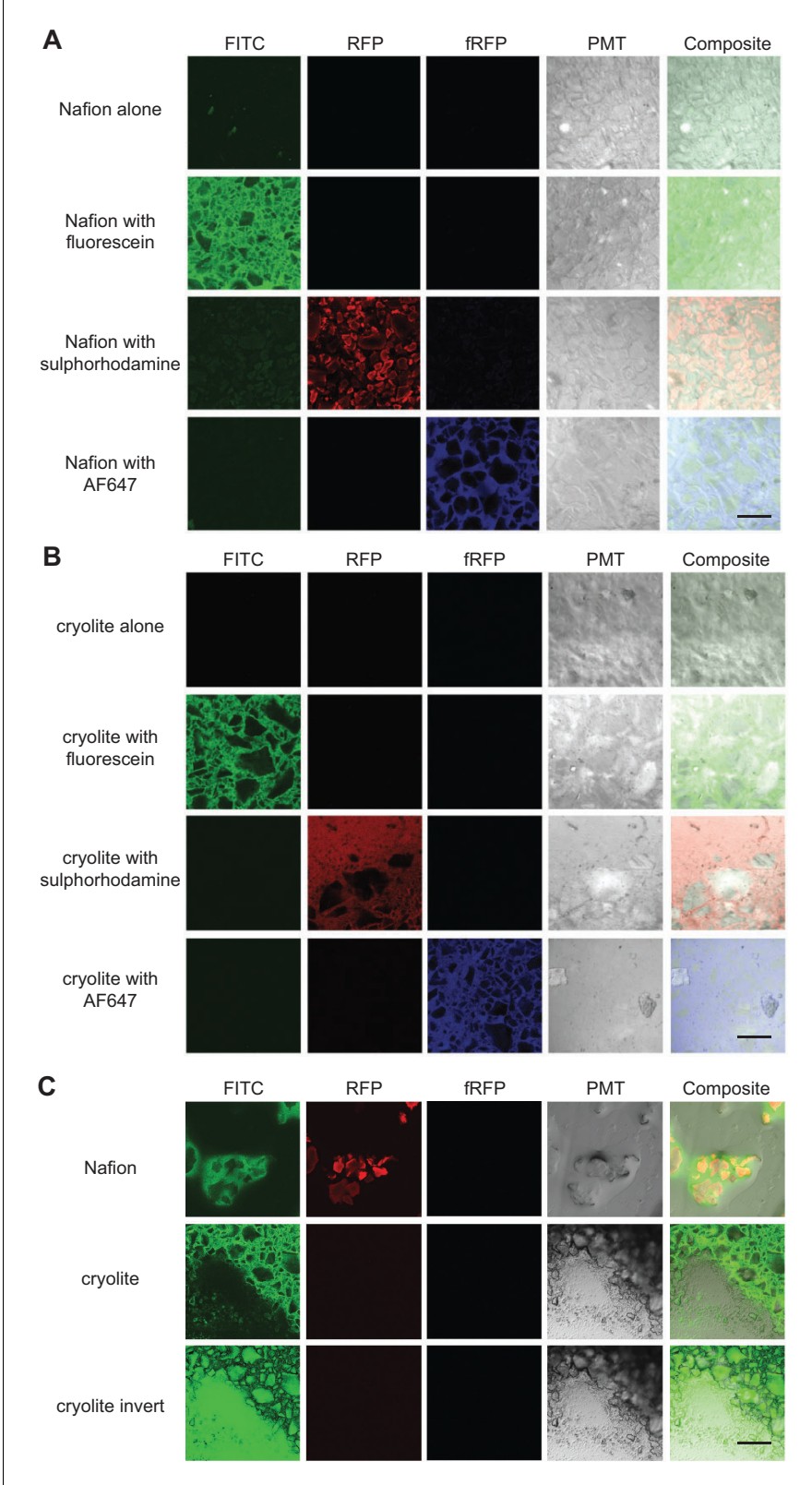

**Figure 2.** Visualization of TS matrices. Nafion (**A**) and cryolite (**B**) were packed into a microwell microscope slide, saturated with water containing the fluorophore indicated, and imaged on a confocal laser scanning microscope with the noted filter sets. A single Z-slice about 10 μm deep into the TS matrix is shown here. 'Cryolite with fluorescein inverted' is the inverted image of 'cryolite with fluorescein', highlighting particles rather than the pore

*Figure 2 continued on next page*

*Figure 2 continued*

space. Partially hydrated microcosms (C) have air-filled pockets (black). Image size of each square is 850 × 850 µm. Scale bars are 250 µm.

The online version of this article includes the following figure supplement(s) for figure 2:

**Figure supplement 1.** Cryolite crystals are invisible under brightfield illumination, but visible under DIC.

**Figure supplement 2.** Particle size distribution of PowdION Nafion powder before and after filtering through 40 µm cell strainer.

autofluorescence was quite weak: the Nafion was less autofluorescent than LB (lysogeny broth), a common microbiological growth medium. Cryolite particle edges, by contrast, are not visible in water by brightfield microscopy, presumably due to close RI matching (*Figure 2—figure supplement 1*). They are visible by DIC microscopy or other polarized light microscopy and are partly visible in the photomultiplier tube (PMT) channel of a confocal microscope (*Figure 2—figure supplement 1*, *Figure 2B*). Cryolite particles were not able to be stained by sulforhodamine, fluorescein, or Alexa Fluor 647 amine dye (*Figure 2B*). They also were not autofluorescent in the FITC, RFP, or fRFP channels (*Figure 2B*). However, the cryolite particle matrix can be visualized by staining the pore water around particles with non-toxic fluorescent dyes such as sulforhodamine, fluorescein, or Alexa Fluor dyes (*Figure 2C*). These images can be inverted to more clearly visualize the particles themselves (*Figure 2C*). Pore water in a Nafion particle matrix can be visualized using these same dyes, which is particularly helpful when imaging unsaturated systems where it is important to be able to distinguish liquid- from air-filled pores (*Figure 2C*). Because cryolite particles cannot be stained and the particles therefore appear as negative space, particles cannot be distinguished from air-filled pores (*Figure 2C*). Nafion is therefore more suitable than cryolite for applications requiring an unsaturated TS matrix.

## Nafion particle size

We tested a range of particle size distributions of TS. We found that, based on their optical properties and our interest in imaging microbes, the optimal particle size distribution was between 2 µm and 40 µm (*Figure 2—figure supplement 2*); we speculate that the particle size distributions using a higher proportion of small (1–2 µm) particles may worsen optical resolution with depth due to light scattering.

## Optical properties of TS microcosms

To assess how well micron-sized objects (e.g. bacteria) could be visualized and resolved within aqueous TS microcosms to 100 µm depth, we mixed 1 µm fluorescent beads within water-saturated TS microcosms and acquired z-stacks from the coverslip up to 100 µm into the microcosms by confocal microscopy (Zeiss 710 CLSM, 40x water immersion objective, NA 1.2, 488/514 nm excitation/emission filters). Bead brightness decreased with depth for both TS substrates, although more steeply for Nafion than for cryolite (*Figures 3A, B and D*). For example, while beads 50 µm into a Nafion microcosm have on average about 40% of the intensity of a bead at the coverslip, beads at the same depth in cryolite only decrease to approximately 60% of their coverslip intensity (*Figure 3A, B and D*). The loss of brightness with depth results in more pixelated images and a lower signal-to-background ratio for 100-µm-deep beads in Nafion compared to 100-µm-deep beads in cryolite (*Figure 3C*). The lower variances between beads deeper in the Nafion matrix are due to the relative dimness of these beads (*Figure 3A*, narrow 95% confidence intervals for beads at 75 and 100 µm). The intensities of the beads are scarcely detectable above the background, and therefore their intensities fall within a compressed range.

To determine the spatial resolution limits of our system, and therefore how suitable it is to imaging of single microbial cells, lateral intensity profiles of 1 µm beads at the coverslip or 100 µm deep were taken and full width at half-maximum (FWHM) values were calculated as a measure of lateral spatial resolution (*Figure 3D and E*). In Nafion, the FWHM of beads 100 µm deep (mean 0.4122 µm ± 0.0086 SD) was greater than the FWHM of beads at the coverslip (mean 0.3755 µm ± 0.0157 SD), indicating an approximately 10% loss of lateral resolution with depth. In cryolite, in contrast, the FWHM (mean 0.3944 µm ± 0.0093 SD) of beads indicated an approximately 5% loss of lateral

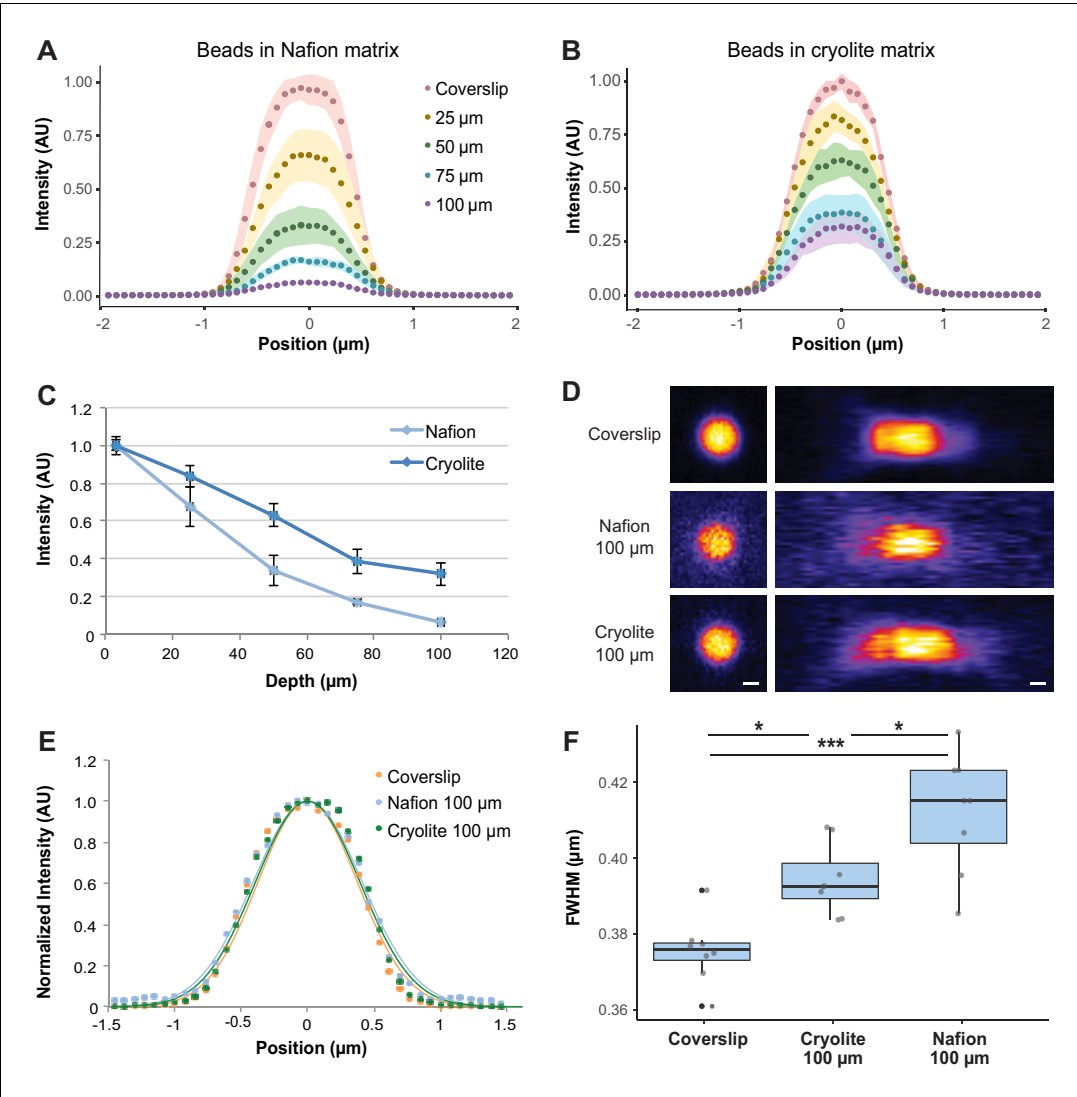

**Figure 3.** Optical properties of TS microcosms. 1 µm FITC-fluorescent beads were mixed into TS matrices saturated with water and imaged by confocal microscopy. Average lateral intensity profiles of beads at different depths within a Nafion (A) or cryolite (B) matrix (n = 8 beads per depth) are shown. (C) Maximum intensities of beads at different depths (n = 8 beads per depth) indicate a greater decay of bead brightness with depth in Nafion than in cryolite. (D) Images of 1 µm beads acquired at the coverslip and 100 µm into Nafion and cryolite microcosms, lateral (xy; left panels) and axial (xz; right panels) views. Image intensity normalized for each image; intensities are not comparable between categories. Scale bar is 0.5 µm. (E) Normalized average lateral intensity profiles of beads at coverslip and at 100 µm depth and fitted Gaussian models. (F) Full-width half-maximum (FWHM) values derived from fitted Gaussian curves of lateral intensity profiles of individual beads are a measure of spatial resolution. FWHM of beads at the coverslip are significantly lower than FWHM of beads 100 µm deep in cryolite (Tukey-Kramer HSD p-value=0.01014) and Nafion microcosms (Tukey-Kramer HSD p-value = $8.6 \times 10^{-6}$), and FWHM for beads in cryolite is significantly lower than beads in Nafion at the same height (Tukey-Kramer HSD p-value=0.0160).

resolution relative to a bead at a coverslip. Overall, beads at 100 µm in cryolite were slightly but significantly better resolved than those in Nafion, with an average difference of about 4% (0.0178 µm) between them (*Figure 3f*; Tukey-Kramer HSD p-value=0.0160; bootstrap p-value testing difference in means = 0.01780). These small losses of resolution at 100 µm depth in both Nafion and cryolite TS microcosms relative to at the coverslip were statistically significant (*Figure 3F*; n = 8 beads per category, one-way ANOVA of all three categories F-statistic 19.904, p-value=$1.418 \times 10^{-16}$; Tukey-Kramer HSD p-value=$8.6 \times 10^{-6}$ for coverslip versus Nafion and p-value 0.01014 for coverslip versus

cryolite; bootstrap p-value testing difference in means 0.0006001 for coverslip versus Nafion and 0.003200 for coverslip versus cryolite). Thus, although previous work focused on closer RI matching of TS particles and liquid in order to improve image quality through the particle matrix (*Downie et al., 2012*; *Leis et al., 2005*; *O'Callaghan et al., 2018*), we found that exact RI matching was not necessary to obtain well-resolved images of 1 μm-diameter fluorescent objects even through 100 μm of TS, suggesting that even micron-sized bacteria could be imaged in both Nafion and cryolite TS microcosms to considerable depth.

## *B. subtilis* bacteria are visible to ~ 75 μm depth in Nafion microcosms and over 100 μm depth in cryolite microcosms

From our assessment of the optical properties of TS microcosms using 1 μm fluorescent beads, we expected that fluorescently labeled bacteria would be resolvable within both TS substrates, but that cryolite would better enable the visualization of bacteria. To test this, we grew *B. subtilis* NCIB3610 expressing a constitutive yellow fluorescent protein (YPet) in aqueous growth medium for two days within TS microcosms, then imaged cells live to 100 μm depth (*Videos 1–4*, *Figure 4*). We chose *B. subtilis* NCIB3610 as the bacterium for this study because it is a model soil saprophyte that can be easily cultured in the laboratory and is genome sequenced and genetically tractable (allowing for the production of fluorescent strains), yet it retains phenotypes lost from many laboratory strains of *B. subtilis* that are important adaptations to life in soils (e.g. biofilm production, secondary metabolite synthesis, etc.; *McLoon et al., 2011*; *Nye et al., 2017*).

As expected, the fluorescence signals from the bacteria and the PMT images both decay with depth. In Nafion, bacteria were visible up to approximately 75 μm depth (*Figure 4A*); in cryolite, bacteria were visible to at least 100 μm depth; we did not image further (*Figure 4B*). Bacteria were more visible in the PMT channel than the YFP fluorescence channel in both Nafion and cryolite (*Videos 1–4*, *Figure 4*). In cryolite in particular, the bacteria were readily visible and identifiable in the PMT channel even at 100 μm, indicating that microbial cells do not necessarily have to be fluorescent in order to be visible in cryolite TS microcosms and suggesting the suitability of the material for non-fluorescently-labeled, non-genetically tractable microorganisms.

## TS microcosms enable submicron resolution imaging of bacteria and fungi in three dimensions over time

Bacteria and fungi are both ecologically important soil microorganisms; we therefore sought to determine whether TS microcosms were compatible with the culture and visualization of both. *B. subtilis* bacteria and *Mucor fragilis* fungi were able to grown and visualized non-destructively over time in both Nafion- and cryolite-based TS microcosms (*Figure 5*). We found that *B. subtilis* cells were still active in hydrated TS microcosms (as indicated by motility within liquid-filled pores) after 2 weeks in MSgg medium (data not shown). *M fragilis* inoculated as spores into TS microcosms grew into mycelial masses within 48 hr (*Figure 5B*), demonstrating the biocompatibility of these materials. Moreover, *M. fragilis* hyphae grown in TS show less variation in width and a more tortuous growth habit (as defined by

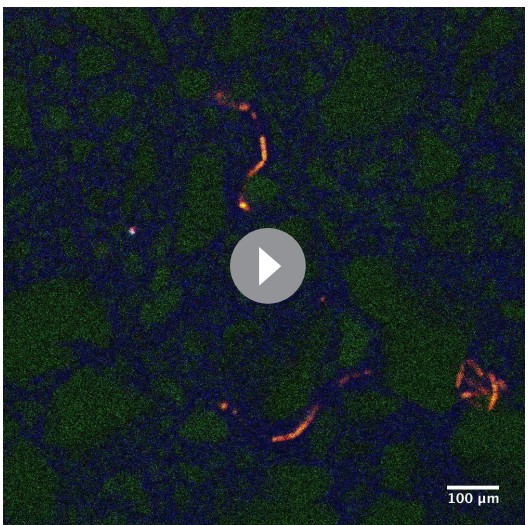

**Video 1.** Z-stack of fluorescently labeled *B. subtilis* 3610 through 100 μm of Nafion (fluorescent channels only). *B. subtilis* 3610 cells expressing constitutive YFP were inoculated into Nafion microcosms in MSgg, incubated at room temperature (22°C) for 48 hr, and Z-stacks acquired by confocal microscopy. Nafion was stained with sulforhodamine, visible in the RFP channel (green), and AF647 amine was added to the liquid medium, visible in the far-red channel (blue). YPet-producing *B. subtilis* cells are visible in the YFP channel (red), with some fluorescence also visible in the RFP channel (green) resulting in yellow appearance. Image size is 101.25 × 101.25 μm. Each frame shows a 0.75 μm slice.
https://elifesciences.org/articles/56275#video1

deviation from straight lines of growth) than *M. fragilis* hyphae are grown without TS (*Figure 5C*). This suggests a possible influence of narrow pore space and touch stimuli offered by particles on fungal branching and morphology (*Soufan et al., 2018*).

The apparent differences in *B. subtilis* growth between Nafion and cryolite microcosms in *Figure 5* are, in our estimation, due to the particular distribution of cells within the small field of view. *Videos 1–4* show a 10x larger field of view than *Figure 5* (and therefore a more representative distribution of cells) at 48 hr. These movies show robust *B. subtilis* growth in both Nafion and cryolite microcosms, indicating comparable abundance and viability.

## TS microcosms are compatible with Raman microspectroscopy and enable in situ single-cell detection of microbial activity as measured by uptake of $D_2O$

All microbial metabolism involves the uptake of water and incorporation of hydrogen from water into cellular biomass. By tracking the incorporation of the non-toxic, isotopically-labeled deuterium from 'heavy' water ($D_2O$) into cellular biomass through peak shifts in the Raman spectrum of single cells, it is possible to detect which cells in a $D_2O$-treated population are active (*Berry et al., 2015*). Peaks in the area between 2030 and 2300 $cm^{-1}$ (referred to hereafter as the 'CD region' or 'CD area', the region indicating Raman signal arising from carbon-deuterium bonds) in single-cell Raman spectra are correlated with $D_2O$ uptake, which in turn is a measure of the metabolic activity of a bacterium (*Berry et al., 2015*).

To test whether it is possible to detect the metabolic activity of single microbial cells in TS microcosms, we grew *B. subtilis* in $H_2O$ medium and in 50% $D_2O$ medium and inoculated them into Nafion and cryolite TS microcosms. (Note that published work shows that most bacteria do not exhibit inhibited growth when cultured in 50% $D_2O$ [*Berry et al., 2015*]). We then obtained Raman spectra from cells embedded 15 to 85 μm deep within the TS matrix. As a control, we also spotted cells onto an aluminum slide and obtained Raman spectra from these cells. (Aluminum slides are a standard Raman microspectroscopy substrate that gives low background signal and allowed us to decouple the Raman spectral variation generated by biological cell-to-cell variability from that generated by the TS matrix).

In agreement with previous work (*Berry et al., 2015*), we found that $D_2O$-grown cells spotted on aluminum slides have a significantly larger CD area than $H_2O$-grown cells (*Figure 6A*; n = 54 for $H_2O$-grown cells, n = 71 for $D_2O$-grown cells; Welch's t-test p-value<$2.2\times10^{-16}$, bootstrap p-value testing difference in means <$1.00\times10^{-7}$). $D_2O$-grown cells show a significantly larger CD area than $H_2O$-grown cells both in Nafion (*Figure 6B*; n = 23 for $D_2O$- and for $H_2O$-grown cells, Welch's t-test p-value=$3.366\times10^{-7}$, bootstrap p-value testing difference in means <$1.00\times10^{-7}$) and in cryolite (*Figure 6C*; n = 41 for $H_2O$-grown cells, n = 26 for $D_2O$-grown cells, Welch's t-test p-value=$1.298\times10^{-5}$, bootstrap p-value testing difference in means <$1.00\times10^{-7}$).

Notably, although $D_2O$-labeled cells can be distinguished from $H_2O$-labeled cells in TS microcosms, the differences between these two populations are about 10-fold lower than when measured on an aluminum slide (note differences in y-axis values for *Figure 6A* versus *Figure 6B and C*). Therefore, in TS microcosms, there is some overlap in these $D_2O$- and $H_2O$-labeled populations, leading to both false-positive and false-negative classifications. In Nafion microcosms, the cutoff for cells to be classified as $D_2O$-labeled can be set at CD region area = 0.5, since all $H_2O$-labeled cells fall below this cutoff (*Figure 6B*). This cutoff produces an approximately 9% false-negative rate ($D_2O$-labeled cells misclassified as $H_2O$-labeled cells; two such cells detected here). In cryolite microcosms, the cutoff for cells to be classified as $D_2O$-labeled was necessarily higher due to the higher background noise from the cryolite matrix (note higher and more varied CD area values for $H_2O$-labeled cells in cryolite versus in Nafion matrix, indicating greater background noise in cryolite). The cutoff for cryolite chambers can be set at CD region area = 0.75, since this is the third quartile of $H_2O$-labeled cells (*Figure 6C*). This cutoff produces a 20% false-positive rate ($H_2O$-labeled cells misclassified as $D_2O$-labeled, due to the eight such cells detected here), and a similar 19% false-negative rate ($D_2O$-labeled cells misclassified as $H_2O$-labeled, due to the five such detected cells).

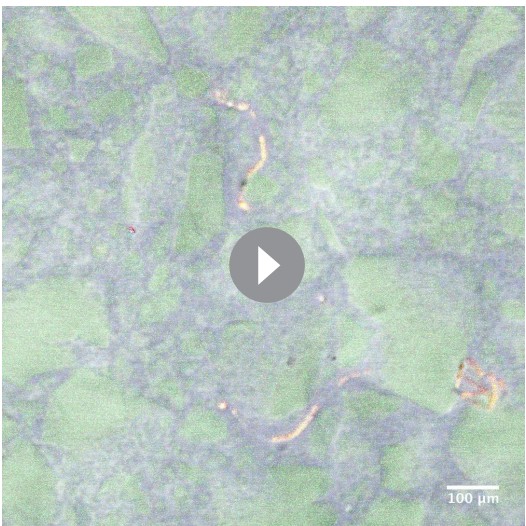

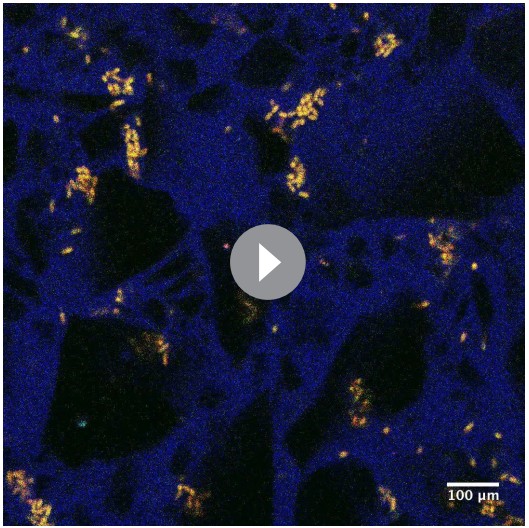

**Video 2.** Z-stack of fluorescently labeled *B. subtilis* 3610 through 100 μm of Nafion (fluorescent channels with PMT channel). Same as in *Video 1*, but with PMT (non-fluorescent) channel added.

https://elifesciences.org/articles/56275#video2

**Video 3.** Z-stack of fluorescently labeled *B. subtilis* 3610 through 100 μm of cryolite (fluorescent channels only). *B. subtilis* 3610 cells expressing constitutive YPet were inoculated into cryolite microcosms in MSgg, incubated at room temperature (22°C) for 48 hr, and Z-stacks acquired by confocal microscopy. AF647 amine was added to the liquid medium, visible in the far-red channel (blue). YPet-producing *B. subtilis* cells are visible in the YFP channel (red), with some fluorescence also visible in the RFP channel (green) resulting in yellow appearance. Image size is 101.25 × 101.25 μm. Each frame shows a 0.75 μm slice.

https://elifesciences.org/articles/56275#video3

## TS microcosms enable the in situ detection of $^{13}$C uptake by bacterial populations by Raman microspectroscopy

Raman spectroscopy is a powerful tool to detect the uptake of $^{13}$C-labeled compounds into cellular biomass by measuring the characteristic 'redshift' of the phenylalanine peak from 1003 cm$^{-1}$ to 966 cm$^{-1}$ upon incorporation of $^{13}$C into the cell (*Huang et al., 2004*; *Huang et al., 2009*; *Li et al., 2012*; *Li et al., 2013*; *Kumar B N et al., 2016*). To test whether we could detect differences in the phenylalanine peak between $^{12}$C- and $^{13}$C-treated cells within TS microcosms, we grew *B. subtilis* cells in growth medium containing either $^{12}$C- or $^{13}$C-glucose and inoculated them into Nafion and cryolite TS microcosms. As a control, we also spotted the cells onto an aluminum slide. We then collected Raman spectra from many single cells within these microcosms.

We quantified $^{13}$C incorporation using the formula $(A_{966}/(A_{966} + A_{1003}))\times100$, where $A_{966}$ represents the area of the $^{13}$C phenylalanine peak and $A_{1003}$ represents the area of the $^{12}$C phenylalanine peak (designated 'Percent $^{13}$C'; see Materials and Methods for details). On aluminum slides, the Percent $^{13}$C metric clearly distinguished $^{12}$C- and $^{13}$C-labeled cells into two non-overlapping populations (*Figure 7A*; Welch's t-test p-value<2.2 × 10$^{-16}$, bootstrap p-value testing difference in means <1.00 × 10$^{-7}$).

In Nafion TS microcosms, however, we found that these differently labeled cells could not be clearly distinguished from each other. Although over three-quarters of $^{13}$C-labeled cells in Nafion had a Percent $^{13}$C value of over 75%, a large number of $^{12}$C-labeled cells also had high Percent $^{13}$C values and were misclassified as $^{13}$C-labeled (*Figure 7B*). As seen in the average spectral traces, $^{12}$C-labeled cells have a high background peak at around 965 cm$^{-1}$ (*Figure 7E*). To determine why this peak occurs even in $^{12}$C-treated cells, we inspected the background spectra of single $^{12}$C-grown cells. A representative background spectrum of a single $^{12}$C-grown cell shows the same peak at 965 cm$^{-1}$ (*Figure 7—figure supplement 1*). This peak within the background may be due to a peak in the same region corresponding to a Nafion-specific C-O-C bond (*Bribes et al., 1991*; *Figure 7—figure supplement 2*). The peak remains in the background-subtracted cell spectrum, resulting in the cell being misclassified as $^{13}$C-labeled. Because with Raman spectroscopy we typically monitor

biomass incorporation of [13]C by measuring spectral shifts in the range of 1003 cm$^{-1}$ to 966 cm$^{-1}$ (which result from isotopic incorporation into the aromatic ring structure of phenylalanine), this background 965 cm$^{-1}$ peak indicates that Nafion may not be ideally suited for microbial carbon-uptake studies. Thus, in this Nafion-based TS microcosm, populations of [13]C-labeled cells did not have a significantly higher Percent [13]C values than [12]C-labeled cells (**Figure 7B**; Welch's $t$-test p-value=0.2151, bootstrap p-value testing difference in means = 0.2288).

In contrast, in cryolite-based TS microcosms, populations of [13]C-treated *B. subtilis* cells could be distinguished from populations of [12]C-treated cells (**Figure 7C**). [13]C-grown cells measured within the cryolite matrix have significantly higher Percent [13]C values than [12]C -grown cells (**Figure 7C**; n = 39 [12]C-grown-cells, n = 20 [13]C-grown cells; Welch's t-test p-value=0.0001348; bootstrap p-value testing difference in means = 0.0002002).

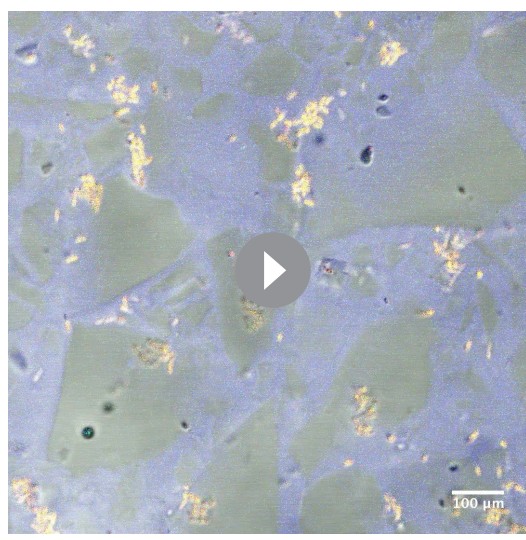

**Video 4.** Z-stack of fluorescently labeled *B. subtilis* 3610 through 100 µm of cryolite (fluorescent channels with PMT channel). Same as in **Video 3**, but with PMT (non-fluorescent) channel added.
https://elifesciences.org/articles/56275#video4

## *B. subtilis* cells attached to dead fungal biomass are more metabolically active after a dry-wet cycle than cells far away from fungal hyphae

Having established that Raman spectroscopy can be used to distinguish single D$_2$O-treated cells from single H$_2$O-treated cells within a Nafion matrix, and knowing that D$_2$O uptake can be used as a marker of microbial activity (**Berry et al., 2015**), we applied our method to an ecologically relevant question: how does a desiccation and rehydration cycle within a porous matrix affect microbial activity, and does proximity to a fungus affect whether cells remain active after being dried?

*M. fragilis* spores were inoculated into Nafion-based TS microcosms and grown overnight in minimal salts glucose growth medium. Fungi were heat-killed and washed in situ. Dead fungi were used instead of live fungi in order to reduce variability and to focus on the effect of the fungal hyphal structure itself on the activity of bacterial cells after a dry-wet cycle – independent of secreted exudates or other potential effects of living fungi. Moreover, fungal necromass may be an important ecological niche for *B. subtilis* in soils: *B. subtilis* strains have been found to exist mainly as spores in soils except when in the presence of dead fungal hyphae, where they were present as vegetative cells (**Siala and Gray, 1974**).

*B. subtilis* cells (three biological replicate cultures) were inoculated into three separate TS microcosms, with the dead *M. fragilis* hyphae as their only carbon source. The microcosm was dessicated for 2 days, then rehydrated with 50% D$_2$O-containing medium as a probe for microbial activity. During washing and rehydration, some cells in the TS microcosms were displaced by liquid flow. We therefore only measured cells that were attached to a surface (either Nafion or *M. fragilis*), reasoning that attached cells would not have moved due to liquid flow. Attachment was measured by attempting to gently pull cells away from their substrate using an optical tweezer (a 1064 nm laser); attached cells did not detach from the substrate when pulled.

We then monitored the deuterium incorporation of *B. subtilis* cells that were classified as being either: on, near (within 20 µm), or far (more than 3 mm) from fungal hyphae (**Figure 8A**). As described above, cells with CD > 0.5 were classified as D$_2$O-labeled. Most *B. subtilis* cells (60% to 70% of cells in each microcosm, n = 3 microcosms) did not show any detectable uptake of D$_2$O after dessication and rehydration (**Figure 8B**, left panel), indicating they were either metabolically inactive (based on the CD detection limit) or no longer viable. However, of the cells that did show uptake of D$_2$O (**Figure 8B**, right panel, cells with CD area >0.5), those attached to *M. fragilis* hyphae incorporated more of the D$_2$O label than did Nafion-attached cells far away from the hyphae (**Figure 8B**;

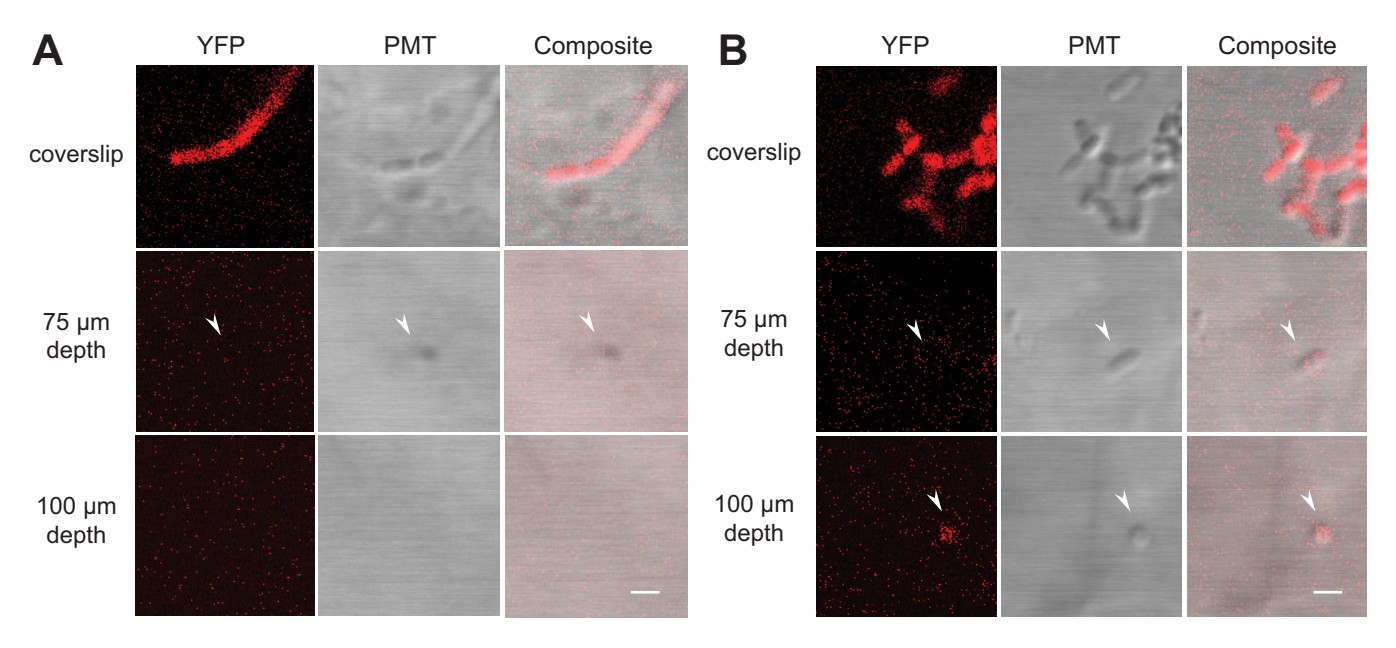

**Figure 4.** Fluorescently labeled bacteria in TS microcosms. *B. subtilis* 3610 cells expressing constitutive YPet were inoculated into TS microcosms in MSgg, incubated at room temperature (22˚C) for 48 hr, and Z-stacks acquired by confocal microscopy. Cells were not fixed, but imaged live. Single slices (0.75 µm thick) acquired at the coverslip, 75 µm, and 100 µm deep into (**A**) Nafion and (**B**) cryolite TS microcosms are shown (frames from *Videos 1*, *2*, *3* and *4*, YFP and DIC channels only). White arrows indicate examples of single visible bacteria. Scale bars = 1 µm.

n = 13 cells on *M. fragilis*, n = 10 cells near *M. fragilis*, n = 11 cells far from *M. fragilis*, one-way ANOVA of all three categories F-statistic 4.5988, p-value=0.01782; Tukey-Kramer HSD p-value=0.01352 for 'on' vs 'far' cells; Welch's t-test p-value for 'on' vs 'far' cells = 0.002398, bootstrap p-value testing difference in means = 0.003600). Cells *near* but not *on M. fragilis* hyphae showed slightly but not significantly higher $D_2O$ uptake than cells far away (Tukey-Kramer HSD p-value=0.4047 for cells near *M. fragilis* versus far from *M. fragilis*).

When cells classified as $D_2O$-labeled (i.e. CD > 0.5) were pooled from three replicate microcosms (with three biological replicate cultures), we found a statistically significant effect of distance category (i.e. 'on', 'near', or 'far' from *M. fragilis*) on CD area (*Figure 8B*; n = 13 cells on *M. fragilis*, n = 10 cells near *M. fragilis*, n = 11 cells far from *M. fragilis*, one-way ANOVA of all three categories F-statistic 4.5988, p-value=0.01782). However, when 'replicate' is added as a factor to the model, neither the replicate nor distance category accounted for the observed variance in CD area (one-way ANOVA of all three categories, distance category p-value=0.2255, replicate p-value=0.2248). Replicates A and B show a trend of decreasing CD area with distance from *M. fragilis*; cells from the replicate C microcosm experiment, however, had such low CD area values overall that only four cells were above the CD > 0.5 detection limit: too few to show a trend with distance category (*Figure 8B*). Therefore, although distance from *M. fragilis* does explain the variation in CD region area when all replicates are pooled, the variation between replicates is high enough that when replicate is added as a factor to the model, the distance category ceases to be significant.

## Discussion

By employing stable isotope probing methods, we were able to detect bacterial activity and carbon uptake within TS microcosms, enabling the non-destructive measurement of the metabolic activity of single cells in a spatially resolved manner. This allowed us to gain insights into an ecologically important process: the microbial response to soil drying and rewetting cycles. This work also establishes the suitability of such microcosms to address additional pressing questions in soil microbial ecology.

In this study, we visualized micrometer-sized bacteria by imaging through TS volumes that represent large bacterial habitats (~100 µm). Although previous studies have used light microscopy to

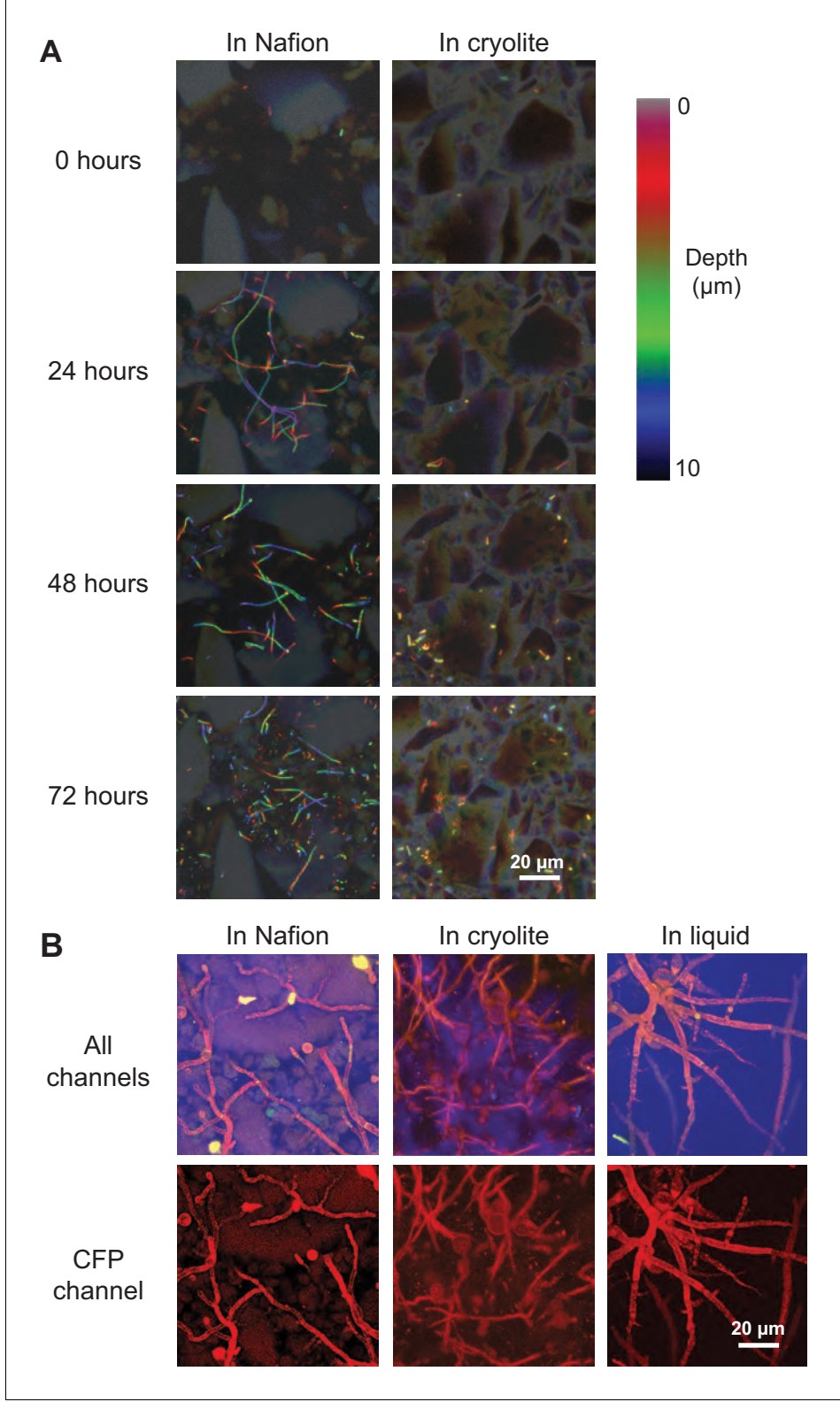

**Figure 5.** Non-destructive imaging of *B. subtilis* and *M. fragilis* in TS over time. (**A**) Cells of a *B. subtilis* 3610 *eps-tasA* biofilm mutant strain expressing constitutive YPet were inoculated into TS microcosms in MSgg, incubated at room temperature (22°C), and imaged over time. Images are ten-micron thick Z-stacks flattened into color-coded Z-projections. Filaments in Nafion (left) are chains of single cells, some of which have segmented, and some of which have not yet expressed autolysins for segmentation and thus appear as filaments (*Chen et al., 2009*). Scale

*Figure 5 continued on next page*

*Figure 5 continued*

bar = 20 µm. (B) *M. fragilis* spores were inoculated into microcosms with or without TS and incubated for 48 hr at 30˚C in MSN minimal salts with 2% glucose. Confocal micrographs of 25 µm Z-stacks flattened into maximum intensity projections; sulforhodamine-stained Nafion (false-colored green), *M. fragilis* autofluorescence in YFP channel (false-colored red), PMT channel (false-colored blue). Scale bar = 20 µm.

image bacteria with single-cell resolution in transparent porous media (*Leis et al., 2005*; *Oates et al., 2005*; *Drescher et al., 2013*), this work further successfully demonstrates the application and combination of light microscopy and Raman microspectroscopy through deep volumes of TS to visualize microbial distributions *and* measure metabolic states. Achieving Raman spectroscopy in TS is a particularly ambitious goal: even with perfect refractive index matching, surface refraction and surface scattering by TS substrates can contribute background noise and decrease the intensity of the Raman signal (*Everall, 2010*; *Freebody et al., 2010*). Despite these challenging conditions, we were able to detect cell-specific Raman spectra from single, micrometer-sized bacteria throughout hydrated TS matrices.

We summarize the properties of both the polymer Nafion and the crystal cryolite for their suitability to soil microbial ecology studies in *Table 1*. Both substrates are non-toxic to heterotrophic bacteria and fungi, are inert (i.e. not readily decomposed); enable the visualization of fluorescent cells through 100 µm depth; have soil-relevant cation exchange properties (empirically measured for Nafion [*Downie et al., 2012*]; theoretically inferred for cryolite [see below]); and are compatible with stable isotope probing by Raman microspectroscopy. However, the two TS substrates have important differences. Nafion has the advantage of being commercially available and pure – as opposed

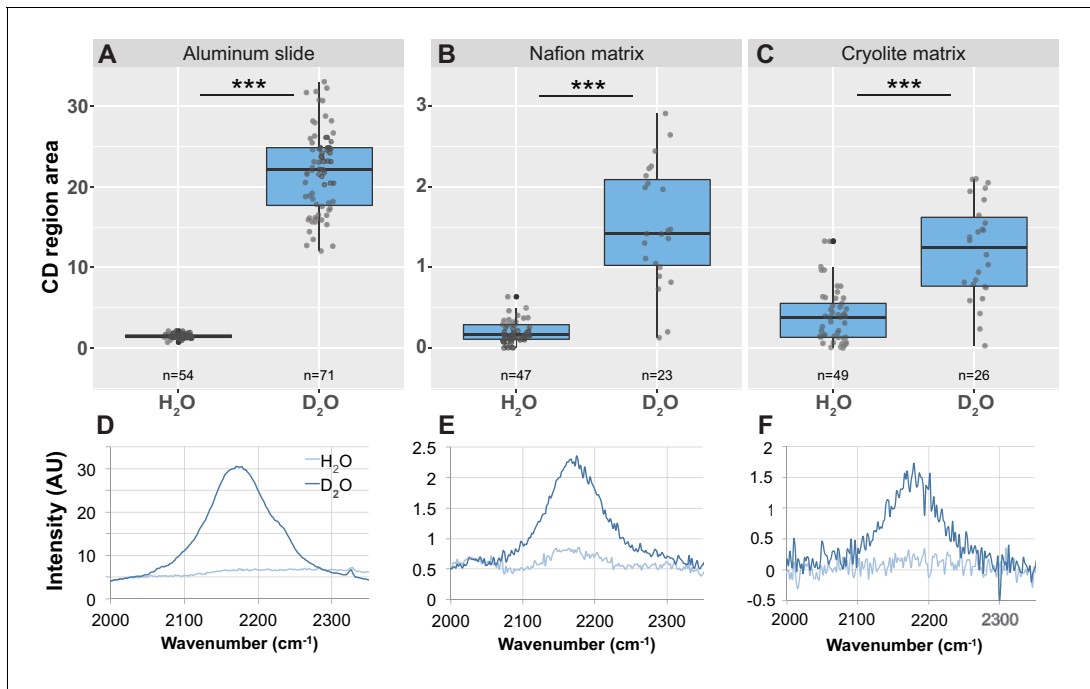

**Figure 6.** Detection of CD peak shifts and $D_2O$ labeling in *B. subtilis* cells in TS microcosms. *B. subtilis* 3610 cells were grown in minimal salts growth medium made with either regular water or 50% heavy water (deuterium oxide, $D_2O$). Cells were then either spotted onto an aluminum slide (A, D) or inoculated into separate Nafion (B, E) or cryolite (C, F) microcosms. Raman spectra of single cells were obtained by microspectroscopy on the aluminum slide, or within the TS matrix from cells embedded anywhere from 15 to 85 µm deep within the matrix. Average background subtracted spectra cells grown in $H_2O$ or $D_2O$ show a broad peak in the CD region of the spectrum between 2050 and 2250 in $D_2O$-labeled cells (D, E, F). CD region for individual cell spectra was calculated as the area under the curve between 2150 and 2200 cm$^{-1}$. Each dot represents CD region for an individual background-subtracted cell spectrum (A, B, C). Each boxplot represents a single separate biological replicate (e.g. all $H_2O$-grown cells inoculated into Nafion came from a single culture, and all $H_2O$-grown cells inoculated into cryolite came from a separate single culture). $D_2O$-labeled cells have larger CD area than $H_2O$-labeled cells on all three substrates (Welch's t-test p-value<1.3 × 10$^{-5}$ for all three substrates).

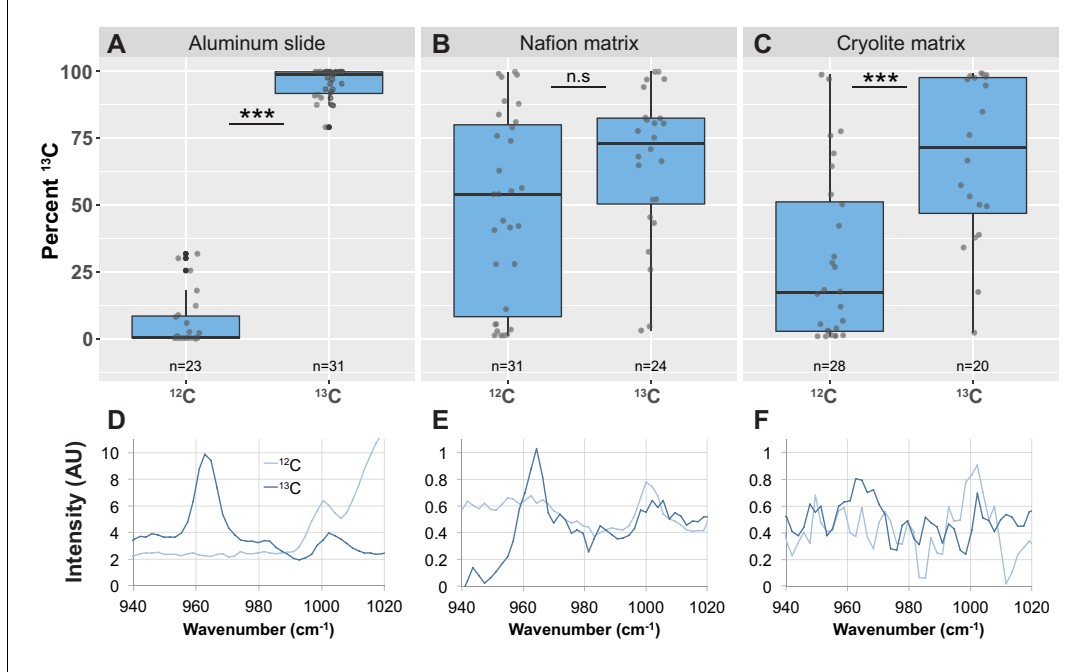

**Figure 7.** Detection of phenylalanine peak shifts and $^{13}C$ enrichment in *B. subtilis* cells in TS microcosms. *B. subtilis* 3610 cells were grown in minimal salts growth medium made with either regular ($^{12}C$) glucose or $^{13}C$ glucose. Cells were then either spotted onto an aluminum slide (A, D) or inoculated into separate Nafion (B, E) or cryolite (C, F) microcosms. Raman spectra of single cells were obtained by microspectroscopy on the aluminum slide, or within the TS matrix from cells embedded anywhere from 15 to 85 μm deep within the matrix. Percent $^{13}C$ is calculated individually for each background subtracted cell spectrum (see Materials and methods). Each dot represents Percent $^{13}C$ for an individual background-subtracted cell spectrum (A, B, C). Each boxplot represents a single separate biological replicate (e.g. all $^{12}C$-grown cells inoculated into Nafion came from a single culture, and all $^{12}C$-grown cells inoculated into cryolite came from a separate single culture). Average background subtracted spectra cells grown in $^{12}C$ or $^{13}C$ glucose are shown (D, E, F). On aluminum slides, $^{13}C$-labeled cells show a significantly higher Percent $^{13}C$ than $^{12}C$-labeled cells (A; Welch's t-test p-value$<2.2\times10^{-16}$). In Nafion TS, the Percent $^{13}C$ of $^{13}C$-labeled cells are not significantly different than $^{12}C$-labeled cells (B; Welch's t-test p-value=0.2151). In cryolite TS, $^{13}C$-labeled cells show a significantly higher Percent $^{13}C$ than $^{12}C$-labeled cells (C; Welch's t-test p-value=$1.348\times10^{-4}$). The online version of this article includes the following figure supplement(s) for figure 7:

**Figure supplement 1.** High Nafion background around 965 cm$^{-1}$ interferes with $^{13}C$ phenylalanine peak at 966 nm, resulting in unlabeled $^{12}C$-rich cells being misclassified as $^{13}C$ labeled.

**Figure supplement 2.** Raman spectra of Nafion and cryolite.

to cryolite crystal, which is sporadically available through gem vendors and may include impurities like siderite, fluorite, and quartz (*Pauly et al., 1999*). It is important to note that synthetically produced cryolite is not recommended as a TS due to the prevalence of sub-micron particles that highly scatter light, the prevalence of opaque aluminum oxides (*Flessa, 1972*), and an amorphous molecular structure that leads to higher baseline Raman spectra (*Tushcel, 2017*).

Cryolite has several advantages over Nafion as a TS substrate. It is considerably cheaper than Nafion (tens of cents per gram as opposed to tens of dollars per gram), readily ground into powder of desired particle sizes by mortar and pestle (as opposed to Nafion, which must be cryomilled or bought pre-ground), and requires no pre-treatment before use (unlike Nafion which requires an extensive number of washes to appropriately functionalize). Cryolite also has the advantage of being extremely optically transparent: micron-sized bacteria can be resolved through even 100 μm of hydrated matrix even under brightfield or DIC illumination on a compound microscope (*Figure 4*, *Figure 2—figure supplement 1*). This makes cryolite an attractive candidate for soil microbial ecology studies, since bacteria may be visualized even without being fluorescently labeled, obviating the need for genetic modification or staining. Cryolite also has more natural dry-down and wet-up behavior than Nafion (which requires small amounts of surfactants [15% ethylene glycol or 0.1% Tween] to be re-wet after drying; although generally non-toxic [*Blank et al., 1987*; *Man et al.,*

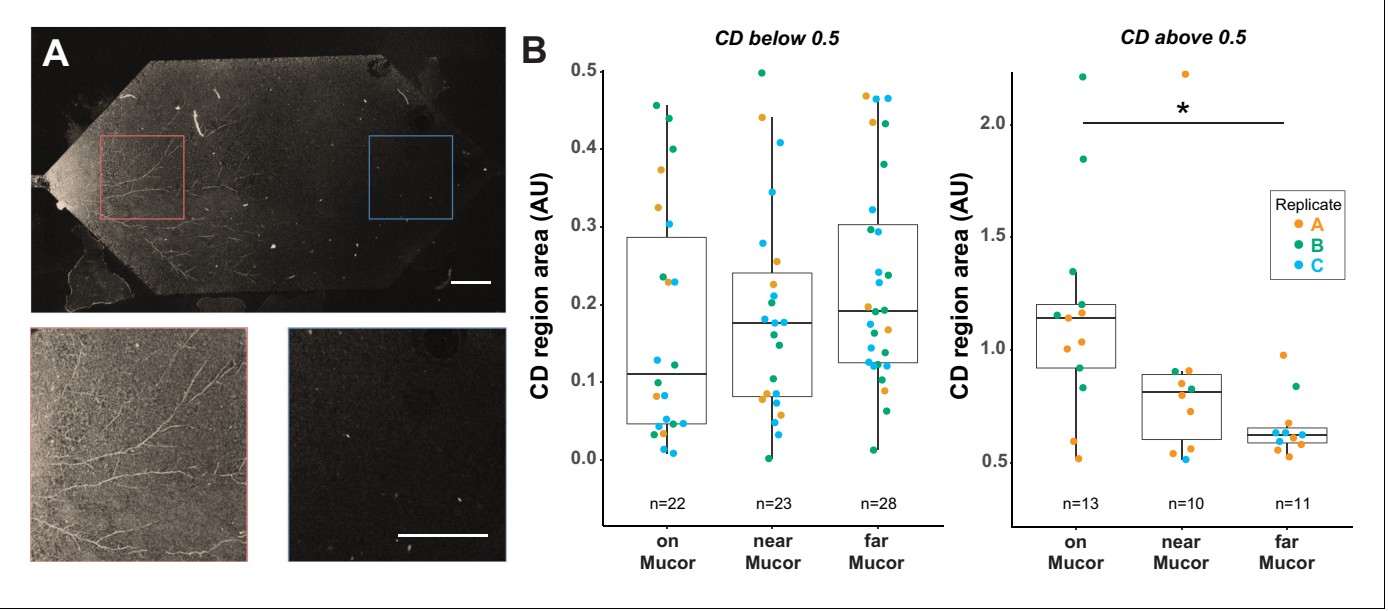

**Figure 8.** *B. subtilis* cells on *M. fragilis* are more metabolically active than cells far from *M. fragilis* after dry-wet cycle in Nafion-based transparent soil microcosms. (**A**) *B. subtilis* cells grown with dead *M. fragilis* in Nafion TS microcosm undergo a dry-wet cycle, mimicking the dry-down and wet-up of soils. Cells were exposed to $D_2O$ for 16 hr during the wet-up phase. Because *M. fragilis* spores were trapped within the Nafion matrix on the side where they were inoculated, after they germinated, one side of the microcosm filled with hyphae, while the other side remains empty. Confocal microscopy in GFP channel shows autofluorescent *M. fragilis* hyphae on one side of microcosm (above, and left inset), and no growth on the other side (right inset). Scale bar is 400 μm in both top micrograph and insets. *B. subtilis* cells measured were classified as either 'on' (cells attached to *M. fragilis* hyphae), 'near' (cells attached to Nafion on *M. fragilis*-inoculated side of microcosm, within 20 μm radius of nearest hypha), or 'far' (cells attached to Nafion on side of microcosm without *M. fragilis*). 'Far' cells are 3 mm or more away from nearest *M. fragilis* hyphae. (**B**) Most cells (~60–70 percent) have no detectable activity after a dry-wet cycle, regardless of distance from *M. fragilis* (left panel, all cells with CD area less than 0.5, indicating no activity detectable by $D_2O$ uptake). However, cells that did take up $D_2O$ took up more of the label when on *M. fragilis* than cells far from *M. fragilis* (right panel, one-way ANOVA of all three categories F-statistic 4.7194, p-value=0.0160; Tukey-Kramer HSD p-value<0.0131 for cells on *M. fragilis* vs far from *M. fragilis,* Welch's t-test p-value=0.002398). Results pooled from three separate biological replicate experiments in three separate TS microcosms, indicated by color.

2017], both of these surfactants can cause cell stress [*Hallsworth et al., 2003*; *Nielsen et al., 2016*] and can be potential sources of nutrient contamination [*Man et al., 2017*]).

An additional consideration for TS materials is their surface charge properties – notably their cation-exchange capacities. After acid washing, the sulfonyl group of Nafion becomes negatively charged with a measured cation exchange capacity of 81 meq per 100 g (*Downie et al., 2012*). Cryolite particles carry a negative charge when in solutions at a pH greater than 1.5 (*Kosmulski, 2009*; *Lindblad and Duroux, 2017*); they are therefore predicted to participate in cation exchange reactions in the pH ranges found in soils (pH 3 to 10; *Slessarev et al., 2016*). For comparison, the point of zero charge for clays are 2.5 for montmorillonite and 4.6 for kaolinite, both of which are important determinants of the cation-exchange capacities of soils (*Kosmulski, 2009*). Thus, both of these TS substrates have cation-exchange capabilities consistent with those expected of natural soils.

These surface properties may also be important to the surprising observed difference in *B. subtilis* 3610 growth habit in Nafion compared to cryolite microcosms (*Figure 4*, *Figure 5*, *Videos 1– 4*). Cells were inoculated as single cells at low initial cell densities into TS microcosms. Cell clusters and filaments (indicating microcolony growth) were visible after 48 hr. In liquid MSgg, about 65% of cells grow into filaments (*Vlamakis et al., 2008*) like the ones visible in Nafion microcosms (*Figure 4A*, *Figure 5A* [left], and *Videos 1* and *2*). The hydrophobic Teflon backbone of Nafion may make it resistant to attachment by *B. subtilis*; the cells therefore grow not on the Nafion particles but largely within the MSgg liquid phase in the microcosm, and thus resemble filaments grown in liquid MSgg. Intriguingly, cryolite appears to induce cell cluster formation in B. subtilis, even in liquid MSgg. The cells localize to cryolite particle surfaces where they form small clusters, not filaments

**Table 1.** Summary table comparing Nafion and cryolite as transparent porous substrates for applications in soil microbial ecology.

| Criterion | Nafion | Cryolite |
|---|---|---|
| Biocompatible? | Yes | Yes |
| Inert (i.e. not easily metabolizable or decomposable by bacteria or fungi)? | Yes | Yes |
| Submicron resolution through 100 µm of matrix? | Yes | Yes |
| Bacteria visible with fluorescence through 100 µm of matrix? | Somewhat (see text) | Yes |
| Bacteria visible without fluorescence through 100 µm of matrix? | No | Yes |
| Ready to use without pre-treatment? | No | Yes |
| Particles visible under brightfield microscopy? | Yes | No |
| Particles visible under DIC and PMT microscopy? | Yes | Yes |
| Particles autofluorescent? | Yes | No |
| Particles visible after staining with fluorescent dye? | Yes | No |
| Pore water visualizable? | Yes | Yes |
| Commercially available? | Yes | Somewhat (see text) |
| Inexpensive? | No | Yes |
| Pure? | Yes | Somewhat (see text) |
| Desired particle size distributions (between 1 µm and ~ 5 mm) easy to obtain? | Somewhat (see text) | Yes |
| Amenable to drying and rewetting? | Somewhat (see text) | Yes |
| Cation-exchanging? | Yes | Yes |
| Compatible with single-cell $D_2O$ tracing by Raman? | Yes | No |
| Compatible with population-level $D_2O$ tracing by Raman? | Yes | Yes |
| Compatible with single-cell $^{13}C$ tracing by Raman? | No | No |
| Compatible with population-level $^{13}C$ tracing by Raman? | No | Yes |

(*Figure 4b*, *Figure 5a* [right], and *Videos 3* and *4*). This clustering may be due to the innate surface properties of cryolite, which is less hydrophobic than Nafion (*Moilanen et al., 2007*). It is an interesting question for future work to determine whether these cryolite-associated cell clusters are physiologically equivalent to bacterial microcolonies that have been observed in native soils (*Grundmann, 2004*; *Nunan et al., 2003*).

We have demonstrated here, for the first time, that it is possible to acquire Raman spectra from cells through both Nafion and cryolite TS matrices. We chose deuterium and $^{13}C$ as stable isotopes because they both produce clear changes in Raman spectra upon cellular uptake, and because they allow questions about microbial activity and carbon metabolism to be explored. $D_2O$-treated cells measured through Nafion were clearly distinguishable from $H_2O$-treated cells using a conservative threshold. Thus, single-cell $D_2O$ uptake measurements can be made using Raman microspectroscopy through Nafion-based TS microcosms. In cryolite-based microcosms, however, variability in the CD area led to a large number (20%) of false positives ($H_2O$-treated cells misclassified as $D_2O$-labeled) (*Figure 6C*). It is therefore impossible to confidently assign a single cell in cryolite as being deuterium labeled. However, at the population level, comparisons of $D_2O$ uptake can be made in cryolite – for example, comparing the CD area of 30 cells exposed to a desiccation and rehydration event versus 30 cells that were not exposed to such an event. This limitation was not observed by *Berry et al., 2015*, whose studies measured the much stronger Raman signals from dried cells on aluminum slides.

*Berry et al., 2015* previously found that an accurate metric for $D_2O$ uptake is the Percent CD, defined as the area of the CD region over the sum of the CD region and the CH region of the cell spectrum. The CH region is the area between 2800 and 3100 cm$^{-1}$, and represents the carbon-

hydrogen bonds in cells, which are depleted in deuterium-labeled cells (*Berry et al., 2015*). In TS microcosms, the CD peak cannot be normalized to the CH region because the measurements are taken in water-filled pores, which produces a large background signal in the CH region due to the Raman spectrum of water itself. In addition, interference from glass and PDMS is high in this region (*Lee et al., 2019*). This may contribute to the greater variability in the CD region area when cells are measured in TS matrices than when $D_2O$-grown cells are measured on aluminum slides (*Berry et al., 2015*).

That said, even without improvements to the optical setup used to acquire Raman spectra within TS microcosms, we expect that substantial gains could be made in cell spectrum quality through improvements in spectral analysis. In this study, our goal was to determine whether any cell signal could be detected over the background of the TS matrix at all. We did indeed detect the cell signal over the background, and used a very simple method to remove the background signal from the cell spectrum: we simply subtracted the Raman spectrum of a nearby point in x-y space from the acquired cell spectrum. However, more sophisticated techniques for removal of defocused light exist and could be applied to improve the quality of spectra obtained from cells within TS matrices, and therefore increase the amount of information that can be derived from these spectra (*Everall, 2010*). For example, nearest-neighbor deblurring methods, previously used to increase the signal of the focal Raman spectrum over the background of a different out- of-focus substrate, could be applied to TS systems (*Govil et al., 1993*). A deconvolution-based removal of background signals based on a model of the contribution of the out-of-focus background region to final cell spectrum may also improve the final results.

Detecting $^{13}C$ uptake through TS matrices by Raman microspectroscopy proved more challenging than detecting deuterium: the signal intensities for the $^{13}C$-associated Raman peaks are weaker than the CD peak (*Huang et al., 2004*; *Berry et al., 2015*; *Eichorst et al., 2015*). Notably, we could not distinguish single $^{13}C$-labeled cells from $^{12}C$-labeled cells in Nafion, which may be due to a Nafion-specific C-O-C bond peak in the same region (*Bribes et al., 1991*); Nafion has other intrinsic Raman peaks in all of the regions that typically indicate $^{13}C$ uptake into cellular biomass (*Figure 7—figure supplement 2*, *Bribes et al., 1991*) since it contains a variety of carbon-based bonds. Cryolite, in contrast, has no carbon bonds, but simply a single repeating bond type arranged in a crystal lattice, resulting in a simple Raman spectrum with a single large peak (Figure S4, *Lafuente et al., 2016*). This low background allows population-level differences in $^{13}C$ uptake to be detected in cryolite TS microcosms.

We used a binary classifier to categorize cells as either labeled or non-labeled, leading to a trade-off between sensitivity (lower false-negative rate) and specificity (lower false-positive rate). The cut-off values for classification can be tuned to favor either sensitivity or specificity, depending on the application. For example, Ho and co-workers used a binary classifier to classify methicillin-resistant and methicillin-susceptible *Staphylococcus aureus* strains based on their Raman spectra (*Ho et al., 2019*). Because the consequences of a false negative (misdiagnosing a harmful resistant strain as a benign susceptible strain) were more severe than the reverse, the authors tuned the classifier to favor greater sensitivity (a lower false-negative rate). Similarly, in using Raman spectra to classify iso-topically labeled and non-labeled organisms, cut-off values should be chosen with the research application and the tradeoff between sensitivity and specificity in mind.

The ability to spatially monitor the distributions and activities of single and populations of cells in soil-like microcosms opens up the possibility of using these TS systems to address long-standing questions in microbial soil ecology. One such question is the role of fungal hyphae in protecting bacterial cells from desiccation and rehydration stress. As soils regularly undergo drying and rewetting events, microorganisms must rapidly adapt to the cellular stress of enormous changes in osmolarity and nutrient availability (*Chen and Alexander, 1973*; *Hallsworth et al., 2003*; *Barnard et al., 2013*; *Kakumanu et al., 2013*). A previous study using NanoSIMS indicated that *B. subtilis* cells take up water, carbon, and nitrogen from nearby fungal hyphae in dry, oligotrophic conditions (*Worrich et al., 2017*). Our Raman microspectroscopy approach is consistent with these results. We found that most *B. subtilis* cells (60% to 70% in each replicate microcosm) do not show any detect-able uptake of $D_2O$ after dessication and rehydration (*Figure 8B*, right panel) indicating their lack of activity. *Berry et al., 2015* were able to detect the CD peak in exponentially growing *E. coli* cells in as soon as 20 min, which is about half of the cell doubling time (47 min), indicating that incorpo-ration of deuterium into macromolecules is detectable well before cell division. In nutrient-replete

liquid shaking cultures, lab strains of *E. coli* and *B. subtilis* have comparable growth rates. We roughly estimate that the cells in our experiment, which were incubated for 16 hr with no detectable signal, had a doubling time over twice as long as the incubation time (therefore greater than 32 hr), indicating that they were either growing extremely slowly relative to replete liquid shaking conditions (over 50 times slower), or were dormant, non-viable, or dead.

The high percentage of non-active cells demonstrates how stressful dry-wet cycles can be, even to bacteria such as *Bacillus* that are considered well-adapted to life in the soil. However, of the cells that showed uptake of $D_2O$, those attached to *M. fragilis* hyphae took up more deuterium than those cells far from *M. fragilis* hyphae (i.e. 3 mm away). Cells *near* but not *on M. fragilis* hyphae – that is cells within 20 μm of the hyphae – showed slightly but not significantly higher $D_2O$ uptake than cells far away. Unlike other single-cell resolution stable isotoping methods like NanoSIMS, Raman microspectroscopy in TS microcosms preserves pore structure since it does not require thin sectioning or dehydration of the sample. Thus, this approach allows measurement of cells not only attached to hyphae, but also those attached to nearby, undisturbed particles.

Improvements in microscopic methods, Raman spectrum analysis, and microcosm reproducibility could be made to further extend the utility of TS microcosms (*Sharma, 2019*). Our method currently has the same limitations as most current microscopy-based methods for investigating environmental microbes: notably, the difficulty of visualizing non-fluorescently tagged, non-genetically tractable microorganisms. Using TS microcosms to study mixed microbial communities therefore presents a significant, but not insurmountable, challenge. While not universally applicable, multiple methods have recently been developed to facilitate the genetic manipulation of wild environmental bacteria (*Brophy et al., 2018*; *Wiles et al., 2018*; *Xu et al., 2020*), which could enable the fluorescent labeling of native bacteria of interest. In the absence of this possibility, morphologically distinct microbes, particularly those of different sizes, could, in principle, be distinguished in TS microcosms. Although, as noted above in *Figure 3D*, slight distortions in the axial plane may make it more difficult to distinguish rod- from sphere-shaped bacteria at deeper depths, future work could potentially use data obtained from oblong beads of varying sizes to develop normalization algorithms to account for such diffraction issues. Vital dyes such as lectin-binding dyes could be perfused gently into the system to stain cells in situ and thereby increase contrast. If fixation is an option for the experiment in question, FISH or other fixed staining approaches could be used.

There also exists the intriguing possibility of conducing 'live' Raman microspectroscopy within the TS microcosms: using Raman spectroscopy not for stable isotope probing (as validated here), but to distinguish and track different microbes non-destructively in situ. There is growing evidence indicating that such strain-specific identifications are possible based solely on the inherent Raman spectra of particular microbes (*Manoharan et al., 1990*; *Harz et al., 2005*; *Rösch et al., 2005*; *Harz et al., 2009*; *Gan et al., 2017*). Indeed, for some strains, Raman spectroscopy can even distinguish cells with distinct physiological states within a single strain – for example, sporulating *Bacillus* cells can be distinguished from vegetative cells by peaks indicating the presence of calcium dipicolinic acid (*Ghiamati et al., 1992*). Obtaining a better view of these Raman signatures may enable us to utilize Raman microspectroscopy to nondestructively monitor bacterial distributions and spatial organizations within TS microcosms. When the field of microbial ecology as a whole is able to routinely use native cell characteristics (such as these Raman signatures or inherent cellular autofluorescence) to identify 'wild' microbes, then this TS microcosm-based method could also be able to be applied towards visualizing native environmental microbes.

In terms of experimental workflow, since the microcosms can be manufactured in batch, the rate-limiting step of utilizing these microcosms (prior to Raman measurements) was the washing steps. The washing and inoculation of microcosms are ideally done on the day of the experiment to maintain sterility, which requires a series of 20 min to 1.5 hr washes (~4 hr of washing total) per microcosm. In addition, since each microcosm requires its own syringe connected to a syringe pump for washing, the number of microcosms that can be prepped simultaneously depends on the number and capacity of syringe pumps available. We had two syringe pumps that handled two syringes each, allowing four microcosms to be washed in parallel. After the washing steps, each microcosm can also be visually checked by microscopy prior to Raman spectroscopy to ensure initial even cell distributions. After inoculation, the rate-limiting step is the manual counting of cells with Raman spectroscopy. A relatively experienced worker can measure about 30 cells with associated background spectra in about 1 to 2 hr.

Further research is needed to definitively assess the reproducibility of TS microcosms. Here, to obtain three replicate microcosms, approximately eight microcosms were initially manufactured. Based on their full particle packing density and strong PDMS-glass seals, six of these were selected for washing; five were retained for inoculation with fungal spores based on their maintenance of high packing density (i.e. minimal loss of particles during washing); four of these were inoculated with bacterial cells based on their well-distributed hyphal growth; and the final three microcosms were used for Raman analysis due to their comparable bacterial cell distributions. For the SIP baseline studies included here (*Figures 6* and *7*), each boxplot represents the variance within one bacterial culture inoculated into one microcosm. Thus, even within a single biological replicate within a single microcosm, the variation between cells is substantial, although the differences between differently labeled cell populations are detectable and significant. In the $D_2O$ uptake experiment after wet-dry cycling (*Figure 8*), each replicate represents a single biological culture within a different microcosm, and there are three such replicates in total. The same trends were not observed in each microcosm; as discussed above, 'replicate' was itself a significant source of variation. Similar heterogeneity and biological variability have been noted even in other, simpler imaging microcosms than those described here. Deng and co-workers, for example, used nearly identical replicate PDMS-based microfluidic devices that mimic the pore properties of soils, and monitored drying of these microcosms under carefully controlled humidity and temperature conditions (*Deng et al., 2015*). Even so, variation between their microcosms was considerable, although major trends were still detected.

Thus, although our transparent soil microcosms are considerably simpler and less heterogenous than true soils – being composed of a single particle type, known particle size distributions, no biological or organic matter at the time of inoculation, and possessing geometric constraints – there remain several sources of heterogeneity between microcosms. The major cause is particle packing; differences in particle density leads to different rates of liquid flow through the microcosms as well as different distributions of cells within the particle matrix. Careful manufacturing technique (filling PDMS chambers as fully as possible with particles before sealing, discarding microcosms that are under- or over-packed) can help mitigate this problem, as well as the use of narrow particle size distributions, and using overall smaller particle sizes (less than 40 μm). In addition to these efforts, running multiple replicate experiments remains important for increasing confidence in findings and the sensitivity of assays. Alternatively, if the heterogeneity cannot be constrained, it can be measured. An entire microcosm can be imaged in 3D by confocal microscopy prior to or after inoculation in order to extract pore size volumes and distributions (for example, see *do Nascimento et al., 2019*). Smaller microcosms and coarser-resolution imaging would enable more rapid imaging of the full volume of the microcosm for this purpose.

In conclusion, the TS model microcosms we describe here – in which the locations, morphologies, growth dynamics, movement, and metabolic states of microbes can be non-destructively resolved – open up the possibility of directly testing many outstanding questions in soil microbial ecology. These include investigations into previously occluded processes such as predation and lysis dynamics, microbial migration patterns, conjugation and transduction networks, and interspecies interactions. The inert nature of the TS substrates described here enables the direct examination of the effects of controlled amounts of nutrients and carbon on soil microbiota. Moreover, extensions of these described TS microcosms with fluorescent in situ hybridization (FISH, to identify microbes and their locations) or fluorescent transcriptional reporter strains (to track bacterial gene expression in native soil-like environments) will further expand the impact of these systems for the study of soil microbial ecology and ecophysiology.

## Materials and methods

**Key resources table**

| Reagent type (species) or resource | Designation | Source or reference | Identifiers | Additional information |
| --- | --- | --- | --- | --- |

*Continued on next page*

*Continued*

| Reagent type (species) or resource | Designation | Source or reference | Identifiers | Additional information |
|---|---|---|---|---|
| Strain, strain background (*Bacillus subtilis*) | NCIB3610 (abbreviated as 3610); ES3 in Shank Laboratory | *Nye et al., 2017* https://doi.org/10.1128/genomeA.00364-17 | GenBank: CP020102.1 | Available from the Shank lab or Bacillus Genetic Stock Center (BGSCID 3A1) |
| Genetic reagent (*Bacillus subtilis* 3610) | ES768 | This study; Shank Laboratory | | *B. subtilis* 3610 *amyE*::P$_{spacC}$-*YPet-cam$^R$*; available from Shank lab |
| Genetic reagent (*Bacillus subtilis* 3610) | ES769 | This study; Shank Laboratory | | *eps-tasA* biofilm gene mutant, *B. subtilis* 3610 *epsA-O::tet tasA::kan amyE*::P$_{spacC}$-*YPet-cam$^R$*; available from Shank lab |
| Gene (*Bacillus subtilis*) | *epsA-0* | GenBank | BSU_34370 through BSU_34220 | Extracellular polysaccharide biosynthesis operon |
| Gene (*Bacillus subtilis*) | *tasA* | GenBank | BSU_24620 | Major component of biofilm matrix |
| Gene (*Bacillus subtilis*) | *YPet* | Ethan Garner, Garner Laboratory | | Yellow fluorescent protein codon-optimized for *B. subtilis* |
| Recombinant DNA reagent | pEA003 | This study; Shank Laboratory | | Available from Shank lab |
| Recombinant DNA reagent | pES037 | This study; Shank Laboratory | | Available from Shank lab |
| Biological sample (*Mucor fragilis*) | | This study; Shank Laboratory | | Isolated the strain from a tall fescue plant in the Piedmont region of North Carolina by Fletcher Halliday, University of North Carolina, 2016; available from Shank lab |
| Software, algorithm | Fiji/ImageJ | https://imagej.nih.gov/ij/docs/guide/146-2.html | RRID:SCR_002285 | |
| Software, algorithm | Raman spectrum analysis software | *Berry et al., 2015*, http://shiny.csb.univie.ac.at:3838/scattr/ | | |
| Other | Nafion | IonPower, Newcastle, DE, USA | POWDion Insoluble −40 + 60 Mesh | https://ion-power.com/product/powdion-insoluble-4060-mesh/ |
| Other | cryolite | Wilhelm Niemetz Minalerien, Vienna, Austria | | Crystalline form from Ivigtut, Greenland |

## Strain construction

*Bacillus subtilis* NCIB3610 strains constitutively expressing fluorescent proteins were constructed by vector cloning, transformation of vectors into *B. subtilis* 168, and phage transduction of *B. subtilis* 168 transformant DNA into *B. subtilis* 3610 (*Yannarell et al., 2019*). *B. subtilis* 3610 is a commonly used strain of *B. subtilis* which is less genetically tractable than *B. subtilis* 168, but exhibits biofilm production phenotypes common to wild *B. subtilis* soil isolates that have been lost in the 168 lab strain (*McLoon et al., 2011*). Plasmid pES037 was constructed by standard restriction digest cloning of *B. subtilis* codon-optimized versions of the *YPet* gene (obtained from Ethan Garner, Harvard University) into the pEA003 parent vector (*amyE*::P$_{spacC}$-*cfp-cam$^R$*), which replaced the *cfp* gene with the new fluorescent protein. Linearized vector was transformed into *B. subtilis* 168 as described previously (*Yannarell et al., 2019*). Cells were plated onto Lennox-chloramphenicol to select for transformants – *amyE*::P$_{spacC}$-*YPet-cam$^R$*. Phage transduction from *B. subtilis* 168 transformants into *B. subtilis* 3610 using SPP1 bacteriophage was performed as previously described (*Yasbin and Young, 1974*), to produce the final strains used in this study: ES768 (*B. subtilis* 3610 *amyE*::P$_{spacC}$-*YPet-cam$^R$*) and ES769 (*eps-tasA* biofilm gene mutant, *B. subtilis* 3610 *epsA-O::tet tasA::kan amyE*::P$_{spacC}$-*YPet-cam$^R$*).

## Microorganisms, media, and growth conditions

*B. subtilis* 3610 strains were routinely cultured on Lysogeny Broth (LB)-Lennox agar (10 g/L tryptone, 5 g/L yeast extract, 5 g/L NaCl, 15 g/L agar) supplemented with antibiotics (5 µg/mL chloramphenicol final concentration) at 30°C for 16–20 hr. *Mucor fragilis* used in this study was obtained from Fletcher Halliday (University of North Carolina at Chapel Hill), who isolated the strain from a tall fescue plant in the Piedmont region of North Carolina, and created a culture stock by patching *M. fragilis* hyphae onto a malt extract agar (MEA) slant (Difco, malt extract 6 g/L, 6 g/L maltose, 6 g/L dextrose, 15 g/L agar) containing 200 µg/mL chloramphenicol. *M. fragilis* was routinely cultured by patching agar from culture stock into the center of an MEA plate containing 200 µg/mL chloramphenicol and incubating at 30°C for 10 days until a lawn of *M. fragilis* spore bodies was obtained. *M. fragilis* spore stock was obtained by adding 10 mL Milli-Q water to this lawn and aspirating the liquid back up to obtain spores. Spores were pelleted by centrifugation (4000 x *g* on tabletop centrifuge in 15 mL Falcon tube) and washed three times in Milli-Q dH$_2$O to remove residual growth medium and hyphae. Spore stock was stored at 4°C for up to 1 month. Spore concentration was calculated by counting spores on hemocytometer and diluting back to 10$^5$ spores per mL for routine use.

MSN medium (minimal salts with free ammonium as nitrogen source; *Beauregard et al., 2013*) was used as the base medium to which various amendments were added for growth and isotope-labeling experiments. MSN is 5 mM potassium phosphate [pH 7], 100 mM morpholinepropanesulfonic acid [MOPS; pH 7], 2 mM MgCl$_2$, 700 µM CaCl$_2$, 50 µM MnCl$_2$, 50 µM FeCl$_3$, 1 µM ZnCl$_2$, 2 µM thiamine, 0.2% NH$_4$Cl. *M. fragilis* was grown in TS microcosms with MSNglu (MSN with 2% glucose, final concentration). *B. subtilis* was grown in TS microcosms with MSNglu or MSgg, a biofilm-inducing medium (5 mM potassium phosphate [pH 7], 100 mM MOPS [pH 7], 2 mM MgCl$_2$, 700 µM CaCl$_2$, 50 µM MnCl$_2$, 50 µM FeCl$_3$, 1 µM ZnCl$_2$, 2 µM thiamine, 0.5% glycerol, 0.5% glutamate). Though the strains used in this study do not metabolize MOPS, it is a carbon-rich molecule that may be metabolized by other microorganisms. A carbon-free buffer, such as phosphate buffered saline, may be substituted for studies requiring a truly carbon-free medium.

## Nafion particle preparation

Nafion is very hydrophobic until its sulfonyl group is functionalized by acid washing, which renders Nafion cation-exchanging and hydrophilic (*Downie et al., 2012*; *Moilanen et al., 2007*). Thus, Nafion particles were acid washed as previously described (*Downie et al., 2012*), with the following modifications. Briefly, 30 g PowdION Nafion powder (IonPower, New Castle, DE, USA) was suspended in 300 mL 15% KOH/35% DMSO aqueous solution in a glass beaker and heated for 5 hr at 80°C to hydrolyze Nafion. Particles were washed three times with Milli-Q water. All washes were carried out by centrifugation unless otherwise noted (4000 x *g* in 50 mL centrifuge tube in tabletop centrifuge for 5 min). To convert Nafion to ion-exchanging form, particles were suspended in 15% nitric acid and incubated at room temperature for 1 hr in a glass beaker, then washed with Milli-Q water, and resuspended with nitric acid and left at room temperature overnight. To remove impurities from Nafion surfaces, particles were washed three times with Milli-Q water, incubated in 1 M sulfuric acid for 1 hr at 65°C in a glass beaker, washed with Milli-Q water, and incubated again at 65°C for 1 hr. Particles were then washed three times with Milli-Q water, suspended in 3% hydrogen peroxide, incubated at 65°C for 1 hr, and washed three times with Milli-Q water. Finally, to replace protons held by Nafion with biologically important cations, particles were washed repeatedly with MSN medium until supernatant pH stabilized at 7.

To obtain particle size fractions small enough to fit in microfluidic microcosms, a dilute particle slurry (1:20 particles to water) was sieved through a 40 µm cell strainer. Particles in the flow-through were collected by centrifugation (see *Figure 2—figure supplement 2* for particle size distributions). Particles were then sterilized by autoclaving slurry (1:1 particles to water by volume).

## Cryolite particle preparation

Cryolite particles were obtained by grinding 5 g of cryolite crystal (Wilhelm Niemutz Mineralien, Vienna, Austria) with a mortar and pestle, washing three times in Milli-Q water by centrifugation, sieving through a 40-µm-cell strainer to obtain small particles in flow-through, and autoclaving particle slurry.

## TS microcosm manufacture and preparation

The TS microcosm construction process is summarized in *Figure 1a*. Sylgard 184 PDMS (Dow Chemical, Midland, MI, USA) was mixed 10:1 base:curing agent, poured over a standard SU-8 silicon master and baked overnight. Inlets and outlets were punched with a 1 mm biopsy puncher (Cole-Parmer, Vernon Hills, IL, USA). Devices were rendered hydrophilic using a handheld corona treater (Electro-Technic, Chicago, IL, USA), passed over the chip 10–20 times at full power. A TS slurry of either Nafion or cryolite particles (1:1 particles to water) was vigorously vortexed and 3 µL of slurry pipetted quickly into PDMS chambers. A glass cover slip (60 × 20×0.15 mm) was corona treated as above and pressed firmly onto the PDMS chamber to seal, then baked at 70°C for 1 hr to bond.

TS in chambers was washed by gently flowing in liquid at a rate of 0.5 µL/min with a syringe pump (Oxford Instruments, Abingdon, UK) using a 1 mL syringe with a 23 gauge needle (BD, Franklin Lakes, NJ) and 0.58 mm ID/0.97 mm OD tubing (Warner Instruments, Hamden, CT). Nafion-based microcosms were rehydrated with 3 uL 20% ethanol to render particles hydrophilic, then washed with 20 µL of desired culture medium before inoculation with microorganisms. Cryolite-based microcosms were simply wet with 3 µL of desired medium (enough to wet the TS) before inoculation (see *Figure 1A*).

## Fluorescent dyes in TS microcosms

Aqueous solutions of sulforhodamine 101, fluorescein (free acid form), and AlexaFluor 647 NHS Ester form dye (AF647) – all from Thermo Fisher (Waltham, MA, USA) – were mixed 1:1 by volume with TS particles and pipetted (40 µL) into ibidi angiogenesis glass bottom slides (ibidi GmbH, Martinsried, Germany) and imaged by confocal microscopy. Sulforhodamine and AF647 were added to 10 µg/mL final volume, and fluorescein was added to 0.1% (w/v) final volume. To dye Nafion particles with sulforhodamine, 10 µg/mL sulforhodamine was added to a 1:1 Nafion:water slurry and incubated at room temperature for 10 min in a roller in order to keep particles shaking. Particles were then washed by settling particles by centrifugation (30 s at 13,200 rcf in a benchtop centrifuge), removing liquid, and adding fresh water or growth medium. For unsaturated microcosms, liquid supernatant was carefully removed from above the settled particles in each well and allowed to air dry partially until about 30% of pore volume was air-filled.

## Fluorescent bead imaging through TS microcosms

1 µm fluorescent beads (Tetraspeck microspheres, Thermo Fisher, Waltham, MA, USA) were mixed into TS microcosms – one Nafion microcosm and one cryolite microcosm. Beads were imaged using a Zeiss 710 confocal laser scanning microscope (Zeiss, Oberkochen, Germany) with a 40x water immersion objective, NA 1.0. 100 µm deep Z-stacks were collected with voxel size of $0.0793 \times 0.0793 \times 0.3535$ µm³. Lateral intensity profiles of beads were measured using Fiji open-source image analysis software (*Schindelin et al., 2012*). Eight beads were measured per measurement depth in each microcosm. Gaussian distribution curves were fitted to each bead and the full width half maximum (FWHM) measured from each fitted curve as a measure of resolution. Maximum intensities of each bead were used as a measure of brightness.

## Bacterial and fungal culture in TS microcosms

*B. subtilis* 3610 cells were grown for 2.5 hr at 37°C shaking in LB broth, washed 5x in appropriate growth medium (MSN or MSgg, see figure captions for details), diluted back to $OD_{600}$ of 0.05 (~$2.5 \times 10^7$ cells/mL) and inoculated into TS microcosms by syringe pump at a rate of 0.5 µL/min, 3 µL total volume. For *Video 1*, *2*, *3*, *4* and *Figure 4*, cells were imaged using a Zeiss 710 confocal laser scanning microscope with a 40x water immersion objective with NA 1.0. 100 µm deep Z-stacks were collected with voxel size of $0.0934 \times 0.0934 \times 0.75$ µm³. For *Figure 5*, cells and fungus were imaged using a Zeiss 880 confocal laser scanning microscope with a 40x water immersion objective with NA 1.0. 10 µm deep Z-stacks were collected with voxel size of $0.0934 \times 0.0934 \times 0.75$ µm³.

*M. fragilis* spores were resuspended 1:10 (from $10^5$ spores/mL spore stock in sterile Milli-Q water) into MSN-glu growth medium and inoculated into TS microcosms by syringe pump. TS microcosms were sealed with Scotch tape on the inlet and outlet and either imaged immediately or placed in a 50 mL Falcon tube with a small piece of wet paper towel to maintain humidity and incubated horizontally at 30°C until removed for imaging.

### *B. subtilis* culture for isotope labeling experiments

*B. subtilis* 3610 cells were grown for 16 to 20 hr at 37°C shaking in MSN 2% glucose medium made with either: 2% regular glucose (termed $^{12}C$ $H_2O$ cells), 2% $^{13}C$ glucose ($^{13}C$ $H_2O$ cells), or 2% regular glucose and 50% (final volume) $D_2O$ ($^{12}C$ $D_2O$ cells). Each cell culture was then inoculated directly into a separate TS microcosms by pipetting ~3 µL cell culture into inlet. In the case of $^{12}C$ $D_2O$ cells, cells were first washed 5x in regular MSN to remove background $D_2O$ (which interferes with Raman cell spectrum signal) and then inoculated into TS microcosms. Cells were measured by Raman microspectroscopy immediately after inoculation.

## Confocal Raman microspectroscopy and spectral processing

Single microbial cell spectra were acquired using a LabRAM HR800 Raman microscope (Horiba, Kyoto, Japan) equipped with a 532 nm neodymium-yttrium aluminium garnet laser and 300 grooves/mm diffraction grating. Spectra were acquired in the range of 400–3200 $cm^{-1}$.

For measurements on aluminum slides, cells were grown, spotted (1 µL), dried, and washed with MilliQ water to remove residual salts. Acquisition parameters were as follows: 30 s acquisition time, 70% laser power, 10% ND filter, slit = 150, hole = 300 (resulting in resolution of approximately 1 µm).

For measurements within TS microcosms, only cells over 15 µm away from either the glass coverslip or PDMS side of the microcosm were measured to reduce interference from glass or PDMS. Single cells were trapped with an optical tweezer (1,064 nm laser; mpc6000; Laser Quantum) during measurement. For Nafion TS microcosms, acquisition parameters were the same as for cells measured on aluminum slides, but with higher laser power (100% versus 70% on aluminum slides) and greater laser transmission (50% ND filter versus 10% on aluminum slides), since cell spectra in TS microcosms are acquired through water which acts as a heat sink and allows heat from a high-laser power to be dissipated before harming or killing cells. For cryolite TS microcosms, laser power (70%) and ND filter (25%) were reduced relative to Nafion TS microcosms, due to unexplained peaks produced in the 966 and 1003 $cm^{-1}$ regions at higher laser powers and intensities.

For each cell spectrum acquired in TS microcosms, a 'background' spectrum in the same z-plane but approximately 10–20 µm away in the x-y plane was acquired as a measure of the Raman spectrum of the local TS matrix around the cell.

Spectra were baselined and normalized using spectrum analysis software (*Berry et al., 2015*). Briefly, each spectrum was fitted with a sixth-degree polynomial function then normalized to total spectral intensity. All spectra collected from a single microcosm were normalized together. The matching background spectrum was then subtracted from each cell spectrum to obtain a background-subtracted cell spectrum for each cell.

'CD region area' for an individual cell spectrum was calculated as the area under the curve between 2145 and 2090 $cm^{-1}$ (average intensity of region between 2145 and 2090 $cm^{-1}$ minus average intensity between the flat baseline region between 2040 and 2060 $cm^{-1}$). 'Percent $^{13}C$' was calculated individually for each background subtracted cell spectrum as $(A_{966}/(A_{966} + A_{1003}))\times100$, where $A_{966}$ is the average intensity between 960 and 968 $cm^{-1}$ minus the average intensity of the nearby baseline 935 to 945 $cm^{-1}$ region ($^{13}C$ phenylalanine peak) and $A_{1003}$ is average intensity between 998 and 1005 $cm^{-1}$ minus the nearby baseline 980 to 985 $cm^{-1}$ region ($^{12}C$ phenylalanine peak).

## Desiccation and rehydration treatment of TS microcosm

*M. fragilis* spores were inoculated into three Nafion TS microcosms with MSN-glu growth medium and incubated at 30°C for 24 hr. Hyphae were heat-killed in situ the next day by incubating the microcosms for 1 hr in a 70°C air incubator, a treatment that kills both hyphae and spores, preventing germination (*Thom et al., 1916*). Microcosms were sealed with Scotch tape during incubation to prevent evaporation. The microcosms were then washed with fresh MSN medium by gently flowing MSN medium through the microcosms at a rate of 0.5 µL/min, 15 µL total volume. *B. subtilis* 3610 cells were then inoculated into the microcosm in MSN medium, with dead *M. fragilis* as their sole carbon source. The fully hydrated microcosm was next dried down by leaving ports on the microfluidics device unsealed and leaving exposed to air in a 50 mL Falcon tube at 30°C for 6 hr. After full desiccation of the Nafion (i.e. macropores filled with air, Nafion reverts to white appearance and not

transparent/translucent as when hydrated), microcosms were incubated for an additional 36 hr at 30°C with ports unsealed and the system exposed to air.

After drying, Nafion becomes very hydrophobic, and cannot be re-wet with water alone. We used 20% ethanol, 15% ethylene glycol, or 0.1% Tween-20 to rehydrate Nafion after drying. 20% ethanol was the fastest and most effective re-wetting agent, and therefore a good choice for initial wetting after microcosm manufacture, when the system is still sterile. However, this high concentration of ethanol is toxic to most cells. We found that 15% ethylene glycol or 0.1% Tween-20 were effective and non-toxic agents (*Blank et al., 1987*; *Man et al., 2017*) for re-wetting Nafion microcosms already inoculated with bacteria or fungi. For our experiments, we therefore slowly percolated 15% ethylene glycol into the microfluidics device using a syringe pump (3 μL total volume, 0.5 μL/min) to rehydrate the hydrophobic Nafion, followed by washing with MSN medium to remove ethylene glycol (20 μL total volume, 0.5 μL/min).

Finally, the system was fully hydrated with MSN made with 50% $D_2O$ (20 μL total volume, 0.5 μL/min) and incubated 16 hr at 30°C, with ports sealed to maintain full hydration. The next day, the microcosm was washed with fresh sterile distilled water to remove excess non-metabolized $D_2O$. Single *B. subtilis* cells were measured by Raman microspectroscopy as described in the previous section. *B. subtilis* cells measured were classified as either 'on' (cells attached to *M. fragilis* hyphae), 'near' (cells attached to Nafion on *M. fragilis*-inoculated side of microcosm, within a 20 μm radius of the nearest hypha), or 'far' (cells attached to Nafion on the side of the microcosm lacking *M. fragilis* hyphae). 'Far' cells are 3 mm or more away from nearest *M. fragilis* hyphae.

## Acknowledgements

We sincerely thank: Carol Arnosti, Alecia Septer, James Umbanhowar, and Peter White for comments on an early draft of the manuscript; Fletcher Halliday (University of North Carolina at Chapel Hill) for the kind gift of the wild *Mucor fragilis* strain used in this study; Ethan Garner (Harvard University) for the *B. subtilis* codon-optimized fluorescent protein gene sequences used in strain construction; Lionel Dupuy (James Hutton Institute) for sharing detailed protocols on Nafion functionalization; the UNC Biology Department Light Microscopy Core facility for use of the Zeiss 710 confocal laser scanning microscope and Tony Purdue for training and assistance on this instrument; the UNC Hooker Light Microscopy Core facility for use of the Zeiss 880 confocal laser scanning microscope and Robert Currin for training and assistance on this instrument; Jamie Winshell (University of North Carolina at Chapel Hill) for assistance in strain construction; and the Division of Microbial Ecology (University of Vienna) for use of their Zeiss AxioObserver epifluorescence microscope, Leica confocal laser scanning microscope, and Horiba high-resolution confocal Raman microspectroscopy system, as well as Markus Schmid for training and assistance on these instruments. This work was supported by funds from the US Department of Energy Office of Biological and Environmental Research Bioimaging Research Program through awards DE-SC0013887 and DE-SC0019012 as well as the European Research Council (Starting Grant: FunKeyGut 741623).

## Additional information

### Funding

| Funder | Grant reference number | Author |
| --- | --- | --- |
| Biological and Environmental Research | DE-SC0013887 | Elizabeth A Shank |
| Biological and Environmental Research | DE-SC0019012 | Elizabeth A Shank |
| European Research Council | FunKeyGut 741623 | David Berry |

The funders had no role in study design, data collection and interpretation, or the decision to submit the work for publication.

## Author contributions
Kriti Sharma, Conceptualization, Data curation, Formal analysis, Validation, Investigation, Visualization, Methodology, Writing - original draft, Writing - review and editing; Márton Palatinszky, Data curation, Formal analysis, Validation, Investigation, Visualization, Methodology, Writing - review and editing; Georgi Nikolov, Formal analysis, Validation, Investigation, Writing - review and editing; David Berry, Conceptualization, Resources, Supervision, Funding acquisition, Validation, Visualization, Methodology, Project administration, Writing - review and editing; Elizabeth A Shank, Conceptualization, Resources, Supervision, Funding acquisition, Visualization, Methodology, Project administration, Writing - review and editing

## Author ORCIDs
Elizabeth A Shank (iD) https://orcid.org/0000-0002-4804-1966

## Decision letter and Author response
Decision letter https://doi.org/10.7554/eLife.56275.sa1
Author response https://doi.org/10.7554/eLife.56275.sa2

# Additional files

## Supplementary files
• Transparent reporting form

## Data availability
All data generated or analyzed during this study are included in the manuscript and supporting files, and source files have been deposited at Dryad.

The following dataset was generated:

| Author(s) | Year | Dataset title | Dataset URL | Database and Identifier |
|-----------|------|---------------|-------------|-------------------------|
| Sharma K, Palatinszky M, Nikolov G, Berry D, Shank EA | 2020 | Transparent soil microcosms enable real-time imaging and non-destructive stable isotope probing of bacteria and fungi | http://dx.doi.org/10.5061/dryad.41ns1rnb3 | Dryad Digital Repository, 10.5061/dryad.41ns1rnb3 |

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
