## [Decision Letter]

**Acceptance summary:**

There is a pressing need in microbial ecology for tools that are reproducible, non-invasive, and mimic native environments while also being compatible with microscopy. In this manuscript, the authors establish and verify protocols for using transparent soils (made of either Nafion or cryolite) to image live microorganisms. This manuscript provides a valuable and timely advance for the microbial ecology community, expanding physical models for measuring bacterial and cross-kingdom interactions in mm-scale communities.

**Decision letter after peer review:**

Thank you for submitting your article "Transparent soil microcosms enable real-time imaging and non-destructive stable isotope probing of bacteria and fungi" for consideration by *eLife*. Your article has been reviewed by three peer reviewers, including Karine A Gibbs as the Reviewing Editor and Reviewer #1, and the evaluation has been overseen by a Reviewing Editor and Gisela Storz as the Senior Editor. The following individuals involved in review of your submission have agreed to reveal their identity: Mette Burmølle (Reviewer #3).

The reviewers have discussed the reviews with one another and the Reviewing Editor has drafted this decision to help you prepare a revised submission.

Summary:

The submission by Sharma et al., (2020) for a Tools and Resources article is well-suited for *eLife*. The authors have provided a critical advance for microbial ecology by establishing and verifying protocols for using transparent soils (made of either Nafion or cryolite) for imaging of live microbes. The reviewers all agree that this research constitutes an important technological advance and that no additional experiments are required. However, each reviewer has noted specific places that need clarification, particularly in some data explanations and in stating the capabilities of the two transparent soil compositions. I encourage the authors to address each of the reviewers' suggestions in the resubmission of this manuscript.

The study by Sharma et al., evaluates previously published, but not regularly or widely used, methods for designing and analysing microbial soil communities. The properties of the two TS microcosms are assessed and compared, and their compatibility with isotope labelling and Raman microspectrometry is evaluated. Microbial ecology is currently in a desperate need of tools, which are reproducible, can be run in parallel, can be analysed by non-constructive sampling (to preserve spatial structure), are compatible with microscopy and, most importantly, mimics conditions faced by microorganisms in the soil. For this reason, I find the study very timely and relevant to a wide span of microbial ecologists.

For many decades, in situ high-resolution microscopy studies in soils have been limited by this medium's incredible high complexity and opacity. This manuscript describes a benchmark study focused on developing two transparent soil matrices (Nafion and cryolite) for micron-scale research applications in microbial species interactions, microbial ecology, and microbial ecophysiology. The authors studied the pros and cons of both matrices in terms of their compatibility with biology (toxicity, pore-spaces, etc), optical and fluorescence microscopy and Raman microspectroscopy. After describing a series of control experiments that lay the basis for an entire new field of studying soil microbes under in situ conditions, they then apply their new platform to studying the anabolic activity of Bacillus cells after a simulated dry-wet cycle. They achieve this by measuring the uptake of deuterium (^2^H) from heavy water (^2^H_2_O) into fatty acids and proteins via single cell resolved and non-destructive Raman microspectroscopy, a comparatively new and yet under-utilized microscopy technique with high potential for the biological sciences.

This study provides the field of (soil and other) microbial ecologists with a new approach to studying microbial physiology and cell-cell interactions under (close to) in situ conditions, at high resolution (sub-micron to micron-scale), and at non-invasive (non-destructive) conditions. This is an exciting new approach with the potential to transform the way soil (sediment etc) microbial ecology studies are performed.

This is one of those very few manuscripts where a reviewer can do nothing but highlight the author's excellent, sound work and can do pretty much nothing else than point out a few ways how the language could be improved (but nothing else). All experiments have been excellently conducted, are well described (please take care of the very few exceptions outlined below), and are scientifically and statistically sound. In contrast to many other studies these days, the authors have done their homework and ran many necessary controls before applying their new technique on a research question. This was a great read!

In the manuscript by Sharma et al., the authors provide methods and further validation of two materials (Nafion and cryolite) for use as transparent soils in microbial ecology. This manuscript provides a valuable service to the community by expanding physical models for measuring bacterial interactions in mm-scale communities, especially for cross-kingdoms interactions. The materials and experimental designs are in an "alpha" version, providing a platform for more innovation and optimization. Table 1 is a nice summary.

Essential revisions:

- Rephrase the title: The title promises too much, as is appears that fungi are also labelled. Perhaps include the aspects of methodology (microcosm) evaluation (which is a large part of the study) in the title, to better reflect the content of the article.

- Throughout the manuscript, clarify the distinction between biological replicates versus samples measured in the same experiment.

- Missing is some discussion on the applicability of the microcosms in regard to mixed bacterial populations. The authors emphasize repeatedly that the microcosms are suitable for analysis of microbial communities and for studying bacterial interactions, but it is not clear to me how this is done, if the bacterial species are non-tagged and morphologically similar (and even if they are not, would the resolution we sufficient to distinguish these at the single cell level (e.g. a coccoid and a rod-shaped)? In my opinion, the study would greatly benefit from including such aspects.

- Please address a lack information on reproducibility of microcosms (although some is provided in the assessment of use of the Raman technology) and number of replicates that can be handled and run at a reasonable effort (i.e. how many microcosms can be handled at the same time in the lab with a 'normal' workload)? In line with this, it could be worth stating, how many biological and technical replicates were analysed to provide data presented and their reproducibility.

- It is not clear how much the autofluorescence of Nafion will interfere with weak fluorescent signals (from e.g. bioreporters or FISH probes).

- Discussion section, it would be a stimulating read to add a few sentences on the potential of live Raman microscopy in TS to the manuscript!

- Subsection “Optical properties of TS microcosms” and throughout: rephrase to "microbial cells" rather than bacterial cells.

- Subsection “Optical properties of TS microcosms”: The diffraction for a presumably circular bead was distorted in the xz, suggesting that rod-shaped organisms would be difficult to resolve fully at 100 µm in either Nafion or cryolite. Future experiments might consider using oblong beads of varying sizes to develop an algorithm or normalization for diffraction at these depths. Also, it would be helpful to know whether the diffraction is impacted by proximity to pores.

- Subsection “TS microcosms are compatible with Raman microspectroscopy and enable *in situ* single-cell detection of microbial activity as measured by uptake of D_2_O”: D_2_O data is not a strong as one would like. The cutoffs seem a bit large for both Nafion and cryolite. Is there a way to control for this in the future?

- Subsection “TS microcosms are compatible with Raman microspectroscopy and enable *in situ* single-cell detection of microbial activity as measured by uptake of D_2_O”: These cut-offs values are ok in the context of your benchmark study but please add a short discussion how more conservative rates for false positive and false negative could be reached by choosing different cut-off values.

- Subsection “*B. subtilis* cells attached to dead fungal biomass are more metabolically active after a dry-wet cycle than cells far away from fungal hyphae”: Did the authors have another way to measure whether cells were attached to surfaces?

- Discussion section: These lines state the most significant insights from this research.

- Discussion section: Please put these results into perspective with the expected growth rate of Bacillus under regular conditions and what the detection limit of deuterium depending on growth rate is; this can be cited away (Berry, 2015 SOI's might have all the info) but it's important to draw the readers’ attention to the fact that longer incubation in presence of heavy water might have changed the conclusion of how many cells are active overall (independent of the rate at which they are active).

- Discussion section: The data does not strongly support the assertion of "single-cell analysis." Rather, analysis of small groups of cells is apparent.

- Subsection “Desiccation and Rehydration Treatment of TS Microcosm”: Discuss why the fungi were killed and how exactly this was done, and whether heating could change the microcosm setup (e.g. if this was dry heat, would this change the water content, etc. or was the chip sealed so evaporation won't take place).

- Subsection “*B. subtilis* cells attached to dead fungal biomass are more metabolically active after a dry-wet cycle than cells far away from fungal hyphae” and aubsection “Desiccation and Rehydration Treatment of TS Microcosm”: The fungi were heat-killed prior to desiccation. Was it considered if fungal spores resisted this heat treatment and if so, how that would affect the results reported?

- Figure 1: Describe the quality control mechanisms. How does one calculate or ensure reproducibility between samples?

- Figure 2: Address the heterogeneity of the pore sizes, especially with cryolite. Is there a mechanism for first scanning the transparent soil in the microfluidic device to describe the pore sizes before starting an experiment?

- Figure 4: Discuss why bacteria are not uniformly seen throughout the matrix.

- Figure 4: Explain the size of the bacterial clusters.

- Figure 4: The authors state that one bacterial cell can be resolved, but this reviewer could not see that in the provided images, even when magnified on-screen. What is the maximum size resolution for the imaging setup?

- Figure 5A: Please discuss the difference in potential *B. subtilis* chaining visible in Nafion vs cryolite.

---

## [Author Response]

Essential revisions:1) Rephrase the title: The title promises too much, as is appears that fungi are also labelled. Perhaps include the aspects of methodology (microcosm) evaluation (which is a large part of the study) in the title, to better reflect the content of the article.

We agree with the reviewers that the initial title was overstated. We have now changed the title to “Transparent soil microcosms for live-cell imaging and non-destructive stable isotope probing of soil microbes”

2) Throughout the manuscript, clarify the distinction between biological replicates versus samples measured in the same experiment.

This clarification has been made throughout.

3) Missing is some discussion on the applicability of the microcosms in regard to mixed bacterial populations. The authors emphasize repeatedly that the microcosms are suitable for analysis of microbial communities and for studying bacterial interactions, but it is not clear to me how this is done, if the bacterial species are non-tagged and morphologically similar (and even if they are not, would the resolution we sufficient to distinguish these at the single cell level (e.g. a coccoid and a rod-shaped)? In my opinion, the study would greatly benefit from including such aspects.

This paragraph has been added to the Discussion section:

“Our method currently has the same limitations as most current microscopy-based methods for investigating environmental microbes: notably, the difficulty of visualizing non-fluorescently-tagged, non-genetically-tractable organisms. Using TS microcosms to study mixed microbial communities therefore presents a significant, but not insurmountable, challenge. While not universally applicable, multiple methods have recently been developed to facilitate the genetic manipulation of wild environmental bacteria (10.1038/s41564-018-0216-5, 10.1128/mBio.01877-18, and 10.1111/1462-2920.15116), which could enable the fluorescent labeling of native bacteria of interest. In the absence of this possibility, morphologically distinct microbes, particularly those of different sizes, could, in principle, be distinguished in TS microcosms (although, as noted above, slight distortions in the axial plane may make it more difficult to distinguish rod- from sphere-shaped bacteria at deeper depths). Vital dyes such as lectin-binding dyes could be perfused gently into the system to stain cells in situ and thereby increase contrast. If fixation is an option for the experiment in question, FISH or other fixed staining approaches could be used.”

4) Please address a lack information on reproducibility of microcosms (although some is provided in the assessment of use of the Raman technology) and number of replicates that can be handled and run at a reasonable effort (i.e. how many microcosms can be handled at the same time in the lab with a 'normal' workload)? In line with this, it could be worth stating, how many biological and technical replicates were analysed to provide data presented and their reproducibility.

The following paragraphs were added to the Discussion section to address these questions of reproducibility and experimental workflow:

“In terms of experimental workflow, since the microcosms can be manufactured in batch, the rate-limiting step of utilizing these microcosms (prior to Raman measurements), was the washing steps. The washing and inoculation of microcosms are ideally done on the day of the experiment to maintain sterility, which requires a series of 20 minute to 1.5 hour washes (~4 hours of washing total) per microcosm. In addition, since each microcosm requires its own syringe connected to a syringe pump for washing, the number of microcosms that can be prepped simultaneously depends on the number and capacity of syringe pumps available. We had two syringe pumps that handled two syringes each, allowing four microcosms to be washed in parallel. After the washing steps, each microcosm must also be visually checked by microscopy prior to Raman spectroscopy to ensure initial even cell distributions. After inoculation, the rate-limiting step is the manual counting of cells with Raman spectroscopy. A relatively experienced worker can measure about 30 cells with associated background spectra in about one to two hours.

Further research is needed to definitively assess the reproducibility of TS microcosms. Here, to obtain three replicate microcosms, approximately eight microcosms were initially manufactured. Based on their full particle packing density and strong PDMS-glass seals, six of these were selected for washing; five were retained for inoculation with fungal spores based on their maintenance of high packing density (i.e. minimal loss of particles during washing); four of these were inoculated with bacterial cells based on their well-distributed hyphal growth; and the final three microcosms were used for Raman analysis due to their comparable bacterial cell distributions. For the SIP baseline studies included here (Figure 6 and Figure 7), each boxplot represents the variance within one bacterial culture inoculated into one microcosm. Thus, even within a single biological replicate within a single microcosm, the variation between cells is substantial, although the differences between differently labeled cell populations are detectable and significant. In the D_2_O uptake experiment after wet-dry cycling (Figure 8), each replicate represents a single biological culture within a different microcosm, and there are three such replicates in total. The same trends were not observed in each microcosm; as discussed above, ‘replicate’ was itself a significant source of variation. Similar heterogeneity and biological variability have been noted even in other, simpler imaging microcosms than those described here. Deng et al., 2015, for example, used nearly identical replicate PDMS-based microfluidic devices that mimic the pore properties of soils, and monitored drying of these microcosms under carefully controlled humidity and temperature conditions. Even so, variation between their microcosms was considerable, although major trends were still detected.

Thus, although our transparent soil microcosms are considerably simpler and less heterogenous than true soils -- being composed of a single particle type, known particle size distributions, no biological or organic matter at the time of inoculation, and possessing geometric constraints -- there remain several sources of heterogeneity between microcosms. The major cause is particle packing; differences in particle density leads to different rates of liquid flow through the microcosms as well as different distributions of cells within the particle matrix. Careful manufacturing technique (filling PDMS chambers as fully as possible with particles before sealing, discarding microcosms that are under- or over-packed) can help mitigate this problem, as well as the use of narrow particle size distributions, and using overall smaller particle sizes (less than 40 μm). In addition to these efforts, running multiple replicate experiments remains important for increasing confidence in findings and the sensitivity of assays.

5) It is not clear how much the autofluorescence of Nafion will interfere with weak fluorescent signals (from e.g. bioreporters or FISH probes).

The following was added (subsection “Visualization of TS matrices”): “For context, the detected autofluorescence was quite weak: the Nafion is less autofluorescent than LB (Lysogeny Broth), a common microbiological growth medium.”

6) Discussion section, it would be a stimulating read to add a few sentences on the potential of live Raman microscopy in TS to the manuscript!

The following section was added to address this question (Discussion section):

“There also exists the intriguing possibility of conducing “live” Raman microspectroscopy within the TS microcosms: using Raman spectroscopy not for stable isotope probing (as validated here), but to distinguish and track different microbes non-destructively in situ. There is growing evidence indicating that such strain-specific identifications are possible based solely on the inherent Raman spectra of particular microbes (Manoharan et al., 1990; Gan et al., 2017; Harz et al., 2005; Harz et al., 2009; Rösch et al., 2005). Indeed, for some strains, Raman spectroscopy can even distinguish cells with distinct physiological states within a single strain – for example, sporulating Bacillus cells can be distinguished from vegetative cells by peaks indicating the presence of calcium dipicolinic acid (Ghiamati et al., 1992). Obtaining a better view of these Raman signatures may enable us to utilize Raman microspectroscopy to nondestructively monitor bacterial distributions and spatial organizations within TS microcosms. When the field of microbial ecology as a whole is able to routinely use native cell characteristics (such as these Raman signatures or inherent cellular autofluorescence) to identify ‘wild’ microbes, then this TS microcosm-based method would also be able to be applied towards visualizing native environmental microbes.”

7) Subsection “Optical properties of TS microcosms” and throughout: rephrase to "microbial cells" rather than bacterial cells.

This change has been made throughout the manuscript except where bacterial cells are being explicitly referenced.

8) Subsection “Optical properties of TS microcosms”: The diffraction for a presumably circular bead was distorted in the xz, suggesting that rod-shaped organisms would be difficult to resolve fully at 100 µm in either Nafion or cryolite. Future experiments might consider using oblong beads of varying sizes to develop an algorithm or normalization for diffraction at these depths. Also, it would be helpful to know whether the diffraction is impacted by proximity to pores.

The following (Discussion section) was added to incorporate these suggestions: “In the absence of this possibility, morphologically distinct microbes, particularly those of different sizes, could, in principle, be distinguished in TS microcosms. Although, as noted above in Figure 3D, slight distortions in the axial plane may make it more difficult to distinguish rod- from sphere-shaped bacteria at deeper depths, future work could potentially use data obtained from oblong beads of varying sizes to develop normalization algorithms to account for such diffraction issues.”

9) Subsection “TS microcosms are compatible with Raman microspectroscopy and enable in situ single-cell detection of microbial activity as measured by uptake of D_2_O”: D_2_O data is not a strong as one would like. The cutoffs seem a bit large for both Nafion and cryolite. Is there a way to control for this in the future?

The following paragraphs were added to the Discussion section.

“Berry and colleagues (2015) previously found that an accurate metric for D_2_O uptake is the Percent CD, defined as the area of the CD region over the sum of the CD region and the CH region of the cell spectrum. The CH region is the area between 2800 and 3100 cm-1, and represents the carbon-hydrogen bonds in cells, which are depleted in deuterium-labeled cells (Berry et al., 2015). In TS microcosms, the CD peak cannot be normalized to the CH region because the measurements are taken in water-filled pores, which produces a large background signal in the CH region due to the Raman spectrum of water itself. In addition, interference from glass and PDMS is high in this region (Lee et al., 2019). This may contribute to the greater variability in the CD region area when cells are measured in TS matrices than when D_2_O-grown cells are measured on aluminium slides (Berry, 2015).

That said, even without improvements to the optical setup used to acquire Raman spectra within TS microcosms, we expect that substantial gains could be made in cell spectrum quality through improvements in spectral analysis. In this study, our goal was to determine whether any cell signal could be detected over the background of the TS matrix at all. We did indeed detect the cell signal over the background, and used a very simple method to remove the background signal from the cell spectrum: we simply subtracted the Raman spectrum of a nearby point in x-y space from the acquired cell spectrum. However, more sophisticated techniques for removal of defocused light exist and could be applied to improve the quality of spectra obtained from cells within TS matrices, and therefore increase the amount of information that can be derived from these spectra (Everall, 2010). For example. nearest-neighbor deblurring methods, previously used to increase the signal of the focal Raman spectrum over the background of a different out- of-focus substrate, could be applied to TS systems (Govil et al., 1991). A deconvolution-based removal of background signals based on a model of the contribution of the out-of-focus background region to final cell spectrum may also improve the final results.”

10) Subsection “TS microcosms are compatible with Raman microspectroscopy and enable in situ single-cell detection of microbial activity as measured by uptake of D_2_O”: These cut-offs values are ok in the context of your benchmark study but please add a short discussion how more conservative rates for false positive and false negative could be reached by choosing different cut-off values.

The following paragraph was added to the Discussion section:

“We used a binary classifier to categorize cells as either labeled or non-labeled, where there is a tradeoff between sensitivity (lower false-negative rate) and specificity (lower false-positive rate). The cut-off values for classification can be tuned to favor either sensitivity or specificity, depending on the application. For example, Ho and co-workers used a binary classifier to classify methicillin-resistant and methicillin-susceptible *Staphylococcus aureus* strains based on their Raman spectra (Ho et al., 2019). Because the consequences of a false negative (misdiagnosing a harmful resistant strain as a benign susceptible strain) were more severe than the reverse, the authors tuned the classifier to favor greater sensitivity (a lower false-negative rate). Similarly, in using Raman spectra to classify isotopically labeled and non-labeled organisms, cut-off values should be chosen with the research application and the tradeoff between sensitivity and specificity in mind.”

11) Subsection “*B. subtilis* cells attached to dead fungal biomass are more metabolically active after a dry-wet cycle than cells far away from fungal hyphae”: Did the authors have another way to measure whether cells were attached to surfaces?

The following line was added (subsection “*B. subtilis* cells attached to dead fungal biomass are more metabolically active after a dry-wet cycle than cells far away from fungal hyphae”): “Attachment was measured by attempting to gently pull cells away from their substrate using an optical tweezer (1,064-nm laser); attached cells did not detach from the substrate when pulled.”

12) Discussion section: These lines state the most significant insights from this research.

We appreciate this insight and moved these sentences to the top of the Discussion section to further highlight them.

13) Discussion section: Please put these results into perspective with the expected growth rate of Bacillus under regular conditions and what the detection limit of deuterium depending on growth rate is; this can be cited away (Berry, 2015 SOI's might have all the info) but it's important to draw the readers’ attention to the fact that longer incubation in presence of heavy water might have changed the conclusion of how many cells are active overall (independent of the rate at which they are active).

The following was added to the Discussion section: “Berry and co-workers (2015) were able to detect the CD peak in exponentially growing *E. coli* cells in as soon as 20 minutes, which is about half of the cell doubling time (47 minutes), indicating that incorporation of deuterium into macromolecules is detectable well before cell division. In nutrient-replete liquid shaking cultures, lab strains of *E. coli* and *B. subtilis* have comparable growth rates. We roughly estimate that the cells in our experiment, which were incubated for 16 hours with no detectable signal, had a doubling time over twice as long as the incubation time (therefore greater than 32 hours), indicating that they were either growing extremely slowly relative to replete liquid shaking conditions (over 50 times slower), or were dormant, non-viable, or dead.”

14) Discussion section: The data does not strongly support the assertion of "single-cell analysis." Rather, analysis of small groups of cells is apparent.

“At the single-cell level” removed from sentence.

15) Subsection “Desiccation and Rehydration Treatment of TS Microcosm”: Discuss why the fungi were killed and how exactly this was done, and whether heating could change the microcosm setup (e.g. if this was dry heat, would this change the water content, etc. or was the chip sealed so evaporation won't take place).

Details on heat-killing protocol were added to Materials and methods section.

In addition, the following lines were added to the Results section: “Dead fungi were used instead of live fungi in order to reduce variability and to focus on the effect of the fungal hyphal structure itself on the activity of bacterial cells after a dry-wet cycle (independent of secreted exudates or other potential effects of living fungi). Moreover, fungal necromass may be an important ecological niche for *B. subtilis* in soils: *B. subtilis* strains have been found to exist mainly as spores in soils except when in the presence of dead fungal hyphae, where they were present as vegetative cells (Siala and Gray, 1974).”

16) Subsection “*B. subtilis* cells attached to dead fungal biomass are more metabolically active after a dry-wet cycle than cells far away from fungal hyphae” and subsection “Desiccation and Rehydration Treatment of TS Microcosm”: The fungi were heat-killed prior to desiccation. Was it considered if fungal spores resisted this heat treatment and if so, how that would affect the results reported?

Fungal spores of Mucor and other zygomycetes are killed at 70C for one hour; citation added (Thom et al., 1916). Details heat-killing protocol were added to Materials and methods section.

17) Figure 1: Describe the quality control mechanisms. How does one calculate or ensure reproducibility between samples?

Quality control and reproducibility are addressed in Point 4 above.

18) Figure 2: Address the heterogeneity of the pore sizes, especially with cryolite. Is there a mechanism for first scanning the transparent soil in the microfluidic device to describe the pore sizes before starting an experiment?

The experiments in Figure 2 were not conducted in PDMS-based TS microcosms, but in microwell slides. This is because the purpose of the experiment was to measure fluorescence and autofluorescence of particles versus background; there was no need to keep cells hydrated over time, and using the pre-manufactured microwell slide was much faster and easier than manufacturing PDMS TS microcosms for each of these experiments. The particle size distribution is therefore looser and less constrained than in a PDMS-based TS microcosm. Though this was mentioned in the Figure 2 caption, we thank the reviewer for catching this oversight. We have clarified this point in more detail in the Materials and methods section as well as by adding this text in subsection “Visualization of TS matrices”:

“(Note that for speed and ease, these fluorescence parameterization experiments were conducted in microwell slides rather than in PDMS microcosms)”

To address the general question of first scanning a microcosm to extract particle size distributions, we have added the following (Discussion section):

“An entire microcosm can be imaged in 3D by confocal microscopy prior to or after inoculation in order to extract pore size volumes and distributions (for example, see Nascimento, 2019, for example). Smaller microcosms and coarser-resolution imaging would enable more rapid imaging of the full volume of the microcosm for this purpose.”

19) Figure 4: Discuss why bacteria are not uniformly seen throughout the matrix.20) Figure 4: Explain the size of the bacterial clusters.21) Figure 5A: Please discuss the difference in potential *B. subtilis* chaining visible in Nafion vs cryolite.

The following paragraph was included in the Results section to address Points 19, 20, and 22:

“These surface properties may also be important to the surprising observed differences in *B. subtilis* growth habit in Nafion compared to cryolite microcosms (Figure 4, Figure 5, Videos). Cells were inoculated as single cells at low initial cell densities into TS microcosms. Cell clusters and filaments (indicating microcolony growth) were visible after 48 hours. In liquid MSgg, about 65% of cells grow into filaments (Vlamakis et al., 2008) like the ones visible in Nafion microcosms (Figure 4A, Figure 5A [left], and Video 1 and Video 2). The hydrophobic Teflon backbone of Nafion may make it resistant to attachment by *B. subtilis*; the cells therefore grow not on the Nafion particles but largely within the MSgg liquid phase in the microcosm, and thus resemble filaments grown in liquid MSgg. Intriguingly, cryolite appears to induce cell cluster formation in *B. subtilis*, even in liquid MSgg. The cells localize to cryolite particle surfaces where they form small clusters, not filaments (Figure 4B, Figure 5A [right], and Video 3 and Video 4). This clustering may be due to the innate surface properties of cryolite, which is more hydrophilic than Nafion.”

22) Figure 4: The authors state that one bacterial cell can be resolved, but this reviewer could not see that in the provided images, even when magnified on-screen. What is the maximum size resolution for the imaging setup?

We changed Figure 4 to show a higher magnification view of the same image where the bacterial cells are more clearly visible.